# A descending inhibitory mechanism of nociception mediated by an evolutionarily conserved neuropeptide system in *Drosophila*

Izumi Oikawa[1], Shu Kondo[2], Kao Hashimoto[3], Akiho Yoshida[1], Megumi Hamajima[4], Hiromu Tanimoto[5], Katsuo Furukubo-Tokunaga[6], Ken Honjo[4,6]*

[1]Graduate School of Life and Environmental Sciences, University of Tsukuba, Tsukuba, Japan; [2]Faculty of Advanced Engineering, Tokyo University of Science, Katsushika-ku, Tokyo, Japan; [3]College of Life and Environmental Sciences, University of Tsukuba, Tsukuba, Japan; [4]Center for Development of Advanced Medicine for Dementia, National Center for Geriatrics and Gerontology, Obu, Japan; [5]Graduate School of Life Sciences, Tohoku University, Sendai, Japan; [6]Faculty of Life and Environmental Sciences, University of Tsukuba, Tsukuba, Japan

**\*For correspondence:**
khonjo@ncgg.go.jp

**Abstract** Nociception is a neural process that animals have developed to avoid potentially tissue-damaging stimuli. While nociception is triggered in the peripheral nervous system, its modulation by the central nervous system is a critical process in mammals, whose dysfunction has been extensively implicated in chronic pain pathogenesis. The peripheral mechanisms of nociception are largely conserved across the animal kingdom. However, it is unclear whether the brain-mediated modulation is also conserved in non-mammalian species. Here, we show that *Drosophila* has a descending inhibitory mechanism of nociception from the brain, mediated by the neuropeptide Drosulfakinin (DSK), a homolog of cholecystokinin (CCK) that plays an important role in the descending control of nociception in mammals. We found that mutants lacking *dsk* or its receptors are hypersensitive to noxious heat. Through a combination of genetic, behavioral, histological, and $Ca^{2+}$ imaging analyses, we subsequently revealed neurons involved in DSK-mediated nociceptive regulation at a single-cell resolution and identified a DSKergic descending neuronal pathway that inhibits nociception. This study provides the first evidence for a descending modulatory mechanism of nociception from the brain in a non-mammalian species that is mediated by the evolutionarily conserved CCK system, raising the possibility that the descending inhibition is an ancient mechanism to regulate nociception.

## eLife assessment

This is a very interesting and **important** study that **convincingly** demonstrates a descending pathway for the control of nociception in non-mammalian organisms.

## Introduction

Minimizing tissue damage is a fundamental task for all animals to increase their chance of survival. Thus, elucidating the principles of nociception, the neural process detecting and encoding potentially tissue-damaging stimuli, is critical to understanding the molecular and neural mechanisms implementing adaptive behaviors and their evolution. Nociceptors are sensory neurons specialized

**eLife digest** Avoiding harm is fundamental for the survival of animals. Nerve cells called nociceptors can detect potential damage, such as extreme temperatures, sharp objects and certain chemicals. In humans, this detection – known as nociception – leads to signals travelling from nociceptors through the spinal cord to the brain, which perceives them as pain.

Mammals such as humans and rodents can inhibit nociception by sending signals from the brain to the spinal cord to dampen pain. This top-down dampening process is believed to play a crucial role in regulating pain in mammals, and it has been implicated in the development of chronic pain. It was not known whether non-mammalian animals shared this inhibitory pathway. However, previous work had shown that fruit fly produce a molecule called Drosulfakinin, which is similar to the chemical that mammals use in the top-down signalling pathway which controls pain.

To determine the role of Drosulfakinin in controlling fly nociception, Oikawa et al. manipulated its activity – and the activity of related genes – in specific neurons in the fruit fly nervous system. Without Drosulfakinin, fly larvae were more sensitive to heat exposure, suggesting that this molecule is required to inhibit nociception. Further experiments showed that Drosulfakinin is present only in the brain of fly larvae and activation of its signaling lowers the activity of neurons that transmit nociceptive signals in the insect equivalent of the spinal cord. This confirms that insect brains can dampen nociception via a top-down pathway, using a similar molecule to mammals.

The findings provide an important foundation for pain studies using non-mammalian animals. The ability to manipulate nociception using genetic techniques in flies offers a powerful tool to understand the top-down process of controlling pain. This result also raises the possibility that this shared top-down inhibition mechanism may have developed over 550 million years ago, which could lead to further research into how nociception and pain regulation systems evolved.

to detect harmful stimuli, whose activation triggers downstream nociceptive circuits and nocifensive responses (*Dubin and Patapoutian, 2010*). Since the activities of nociceptors and downstream nociceptive circuits are tightly linked to pain perception in humans, unveiling the mechanisms of nociception is also crucial to a better understanding of human pain mechanisms (*Burrell, 2017*; *Walters, 2018*).

Descending inhibition has been suggested to be a pivotal mechanism in the modulation of nociception and pain in mammals. Since the discovery that electrical stimulations of parts of the midbrain in rats enabled surgical operations without anesthetics (*Reynolds, 1969*), mammalian descending nociceptive pathways have been implicated in various analgesic phenomena/treatments and the development of chronic pain states, suggesting their critical role in modulating nociception and pain (*Chen and Heinricher, 2019*; *Ossipov et al., 2014*). However, brain-mediated modulatory mechanisms of nociception such as descending inhibition have currently been identified only in mammals, despite a high degree of commonality across species in the peripheral nociceptive mechanisms (*Arenas et al., 2017*; *Sneddon, 2018*). Therefore, it is unknown whether the descending modulation is a de novo mechanism typical of the highly developed mammalian central nervous system (CNS) or a conserved control also present in simpler animals.

The descending nociceptive-modulatory systems in mammals have been revealed to involve various neurochemical pathways (*Chen and Heinricher, 2019*; *Ossipov et al., 2014*; *Ossipov et al., 2010*); among these, the cholecystokinin (CCK) system is one of the most extensively characterized (*Ossipov et al., 2014*; *Ossipov et al., 2010*). In rodents, CCK signaling plays a crucial role in facilitating nociception by counteracting the opioidergic systems in the periaqueductal gray (PAG)—rostral ventral medulla (RVM)—spinal descending pathway (*Heinricher and Neubert, 2004*; *Heinricher et al., 2001*; *Marshall et al., 2012*; *Xie et al., 2005*) and inhibiting nociception through the central amygdala (CeA)—PAG—spinal pathway (*Roca-Lapirot et al., 2019*). In humans, CCK signaling has been implicated in nocebo hyperalgesia, mediated by the descending nociceptive control system (*Manaï et al., 2019*). The CCK system is very well-conserved and has been implicated in several common physiological functions among bilaterian species (*Elphick et al., 2018*; *Mirabeau and Joly, 2013*; *Jékely, 2013*; *Nässel and Williams, 2014*; *Tinoco et al., 2021*). However, whether CCK is functionally involved in regulating nociception outside of mammals remains unknown.

Drosulfakinin (DSK), a neuropeptide homologous to CCK, was identified in the fruit fly *Drosophila melanogaster* (*Nässel and Williams, 2014*). Fly DSK is reportedly involved in modulating many physiological functions shared with mammals, including gut functions, anxiety, aggression, memory, feeding, synaptic functions, and courtship behaviors (*Nässel and Williams, 2014*; *Wu et al., 2019*; *Wu et al., 2020*; *Mohammad et al., 2016*). After the discovery of stereotyped nociceptive escape behavior called rolling and polymodal Class IV md (C4da) nociceptors (*Tracey et al., 2003*; *Hwang et al., 2007*), the larval *Drosophila* has been successfully utilized to identify evolutionarily conserved and previously uncharacterized molecular pathways in nociception (*Zhong et al., 2012*; *Kim et al., 2012*; *Zhong et al., 2010*; *Mauthner et al., 2014*; *Babcock et al., 2011*; *Honjo et al., 2016*; *Honjo and Tracey, 2018*; *Neely et al., 2010*; *Follansbee et al., 2017*) and circuitry mechanisms in the ventral nerve cord (VNC; the invertebrate equivalent of the spinal cord) to compute multimodal sensory stimuli and select nociceptive escape strategies (*Burgos et al., 2018*; *Kaneko et al., 2017*; *Takagi et al., 2017*; *Dason et al., 2020*; *Hu et al., 2020*; *Chin and Tracey, 2017*). Previous studies have demonstrated that neuropeptidergic systems also participate in regulating nociception in *Drosophila* (*Hu et al., 2020*; *Hu et al., 2017*; *Im et al., 2015*; *Bachtel et al., 2018*; *Aldrich et al., 2010*). However, the role of fly DSK in nociception remains elusive.

Here, using a collective approach of genetic, behavioral, histological, and $Ca^{2+}$ imaging analyses, we pursued the mechanisms of DSK-mediated nociceptive regulation and demonstrate that the DSK system constitutes a descending inhibitory pathway of nociception from the brain to the VNC in larval *Drosophila*.

## Results

### DSK signaling negatively regulates thermal nociception

Through a thermal nociception screen using the nocifensive rolling response of *Drosophila* larvae, we found that a deletion mutant line of the *dsk* gene showed thermal hypersensitivity with a significantly shorter latency in their response to a 42 °C probe than the controls, suggesting that DSK plays a role in negatively regulating nociception (*Figure 1A and B*). A genomic fragment containing the wild-type *dsk* gene, and no other neuropeptide genes, significantly rescued the thermal hypersensitivity of the *dsk* mutants (*Figure 1B*), confirming that *dsk* is responsible for the thermal hypersensitivity.

DSK has been shown to activate two G-protein coupled receptors, CCKLR-17D1 and CCKLR-17D3 (*Chen et al., 2012*; *Kubiak et al., 2002*), which are orthologous to the mammalian CCK receptors, CCKAR (also known as $CCK_1$) and CCKBR (also known as $CCK_2$) (*Elphick et al., 2018*; *Mirabeau and Joly, 2013*; *Jékely, 2013*). To test whether these receptors mediate DSK signaling in nociception, we generated deletion mutants for *CCKLR-17D1* and *CCKLR-17D3* using CRISPR/Cas9 genome editing and tested them for thermal nociception (*Figure 1C–F*). When stimulated with a 42 °C probe, three independent deletion lines of the *CCKLR-17D1* and *CCKLR-17D3* exhibited thermal hypersensitivity (*Figure 1D and F*), further supporting the role of DSK signaling in the negative regulation of thermal nociception. To examine whether the phenotypes of *CCKLR-17D1* and *CCKLR-17D3* mutants are classified as hyperalgesia (hypersensitivity to normally noxious stimuli) or allodynia (abnormal hypersensitivity to normally innocuous stimuli), we tested the receptor mutants with a 38 °C probe, which is close to the threshold of larval thermal nociception (39 °C; *Tracey et al., 2003*). We found that the responses of the DSK receptor mutants were indistinguishable from the controls, indicating that the DSK receptor mutants are hypersensitive to suprathreshold thermal stimuli, thus hyperalgesic (*Figure 1—figure supplement 1*).

We have noticed that the *yw* control strain, which was used by us to generate the *dsk* and receptor deletion mutants, showed relatively longer response latencies to the 42 °C probe compared to the other control strains used in this study. This may be attributed to the effect of the genetic background, although, presently, the cause for this difference is unknown.

### Two groups of brain neurons expressing DSK are responsible for regulating nociception

We attempted to identify DSK-expressing cells in the larval CNS that are responsible for regulating nociception. Unlike mammalian CCK, which is expressed in the CNS and gastrointestinal system, *Drosophila* DSK is expressed in the CNS but not in the gut (*Nässel and Williams, 2014*; *Nichols and*

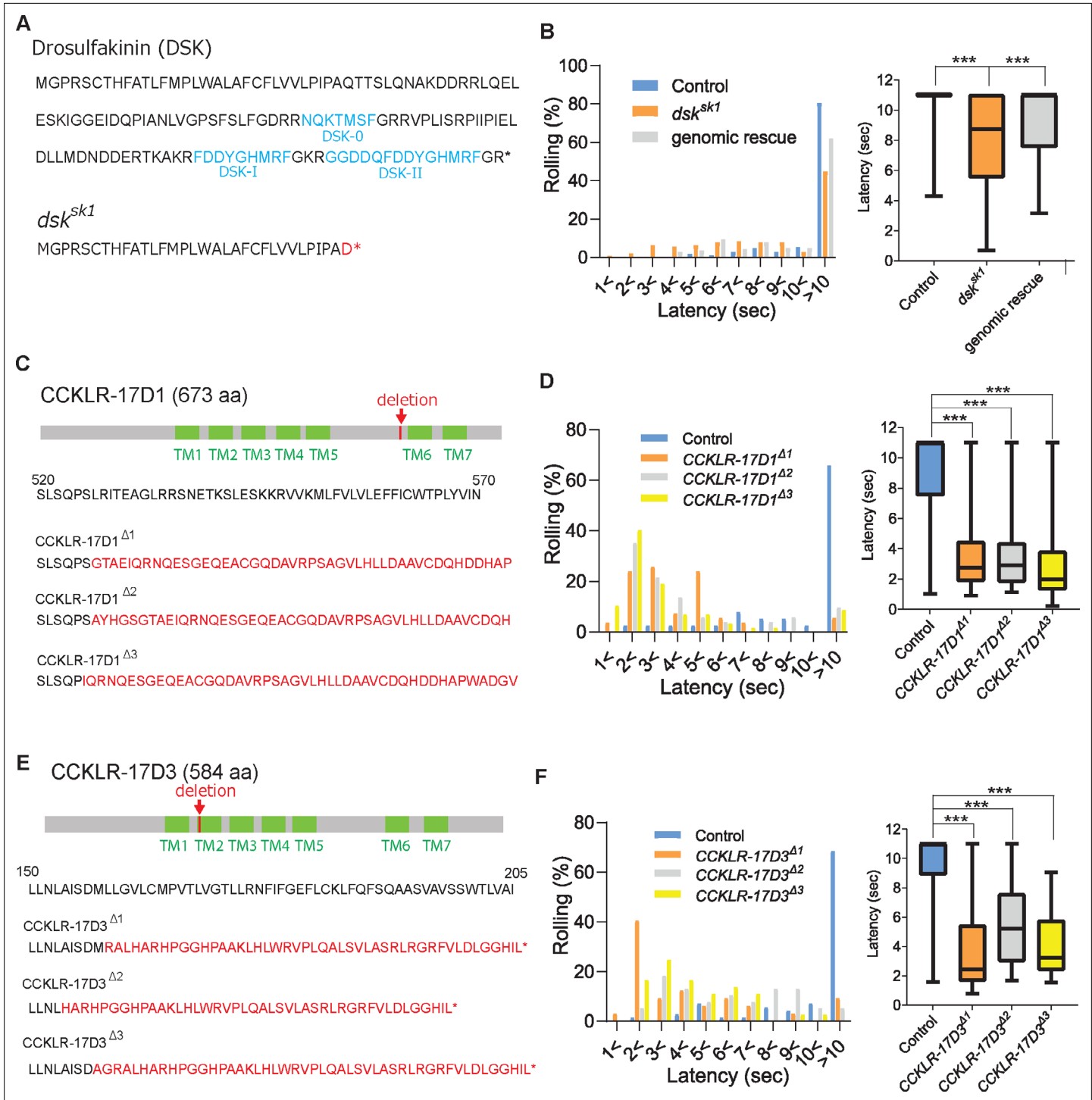

**Figure 1.** DSK signaling is involved in negatively regulating nociception. (**A**) Predicted amino acid sequences of pro-DSK peptide in the wild-type (top) and *dsk*[sk1] mutants (bottom). Due to a 5-base deletion at the +94–99 position in the coding sequence, *dsk*[sk1] mutants are predicted to produce a largely truncated pro-DSK peptide unable to be processed to active DSK peptides. Red letters represent residues different from the wild-type and asterisks indicate stop codons. (**B**) Hypersensitivity of *dsk*[sk1] mutants to a 42 °C thermal probe. Significantly shortened latencies of *dsk*[sk1] mutants (n=143) compared with the controls (n=164) were recovered in genomic rescue animals (*Dp(3;1)2-2; dsk*[sk1], n=139). (Left) Histograms (Right) Box plots of latencies. *** p<0.001 Steel's test. (**C**) Predicted amino acid sequence of CCKLR-17D1 in the wild-type and *CCKLR-17D1* mutants. *CCKLR-17D1*[Δ1], *CCKLR-17D1*[Δ2], and *CCKLR-17D1*[Δ3] have 17-base (+1573–1589), 2-base (+1575–1576) and 32-base (+1573–1604) deletions in the coding sequence respectively, which result in large frameshifts completely abolishing the sixth and seventh transmembrane domains (TMs). (**D**) *CCKLR-17D1*[Δ1] (n=54), *CCKLR-17D1*[Δ2] (n=51) and *CCKLR-17D1*[Δ3] mutants (n=57) all showed significantly shorter latencies than control (n=38). (Left) Histograms (Right) Box

*Figure 1 continued on next page*

*Figure 1 continued*

plots of latencies. *** p<0.001 Steel's test. (**E**) Predicted amino acid sequence of CCKLR-17D3 in the wild-type and *CCKLR-17D3* mutants. *CCKLR-17D3^{Δ1}, CCKLR-17D3^{Δ2}*, and *CCKLR-17D3^{Δ3}* possess 8-base (+474–481), 32-base (+457–488), and 5-base (+472–476) deletions in the coding sequence respectively, which result in large frameshifts and truncation in the middle of the second TM. (**F**) *CCKLR-17D3^{Δ1}* (n=32), *CCKLR-17D3^{Δ2}* (n=38), and *CCKLR-17D3^{Δ3}* mutants (n=36) all exhibited significantly shorter latencies than control (n=70). (Left) Histograms (Right) Box plots of latencies. *** p<0.001 Steel's test. All box plots show median (middle line) and 25th to 75th percentiles with whiskers indicating the smallest to the largest data points.

The online version of this article includes the following source data and figure supplement(s) for figure 1:

**Source data 1.** Source data used to generate the summary data and graphs shown in *Figure 1* and *Figure 1—figure supplement 1*.

**Figure supplement 1.** Thermal nociceptive thresholds in DSK receptor mutants are largely normal.

*Lim, 1996*; *Veenstra et al., 2008*; *Söderberg et al., 2012*; *Park et al., 2008*). Previous immunohistochemical studies have reported putative DSK-expressing cells in the larval CNS (*Nichols and Lim, 1996*; *Söderberg et al., 2012*). However, since the specificity of the DSK antibodies has not been validated with null mutants in the former studies, it has been a concern that the reported DSK-expressing cells could include non-DSK cells that express the other neuropeptides sharing the C-terminal RFa motif with DSK (*Nässel and Winther, 2010*). Consistent with this concern, we found that an antibody against crustacean FLRFa, an FMRFa-like neuropeptide with the C-terminal RFa motif, gives rise to a comparable staining pattern to that of the previously reported DSK antibodies, visualizing cells designated as insulin-producing cells (IPCs; referred to as SP3 in *Nichols and Lim, 1996*), MP1, SP1, SP2, Sv, SE2, Tv1-3, T2dm, and A8 in the larval CNS (*Figure 2A*; *Nichols and Lim, 1996*; *Söderberg et al., 2012*).

To identify bona fide DSK-expressing cells responsible for regulating nociception, we performed anti-FLRFa staining in multiple *dsk* null alleles and found that the staining signals persist in all but two pairs of neurons, MP1 and Sv, in the mutant CNS (*Figure 2B–D* and *Figure 2—figure supplement 1A–D*), suggesting that these two pairs of neurons are the only neurons in the larval CNS that express DSK. The genomic rescue fragment that rescued the thermal hypersensitivity of *dsk* mutants (*Figure 1B*) also restored the anti-FLRFa signals in MP1 and Sv in a *dsk* mutant background (*Figure 2E*), suggesting that the *dsk* gene is responsible for the anti-FLRFa signals in these neurons as well as the thermal nociceptive responses of larvae. This expression pattern was further corroborated by the transgenic reporter line *DSK-GAL4* and the 2A-GAL4 knock-in reporter line *DSK-2A-GAL4* (*Deng et al., 2019*), both of which we found were expressed only in MP1 and Sv neurons among the anti-FLRFa-positive neurons (*Figure 2A, C and F*; *Figure 2—figure supplement 1B–F*). We also found that anti-FLRFa, *DSK-GAL4*, and *DSK-2A-GAL4* do not visualize larval peripheral neurons including C4da nociceptors (*Figure 2A and F*; *Figure 2—figure supplement 1G*). Taken together, these results identified two sets of brain neurons, MP1 and Sv, as the DSK-expressing cells that are involved in regulating nociception in larvae.

## CCKLR-17D1 in Goro neurons functions to negatively regulate nociception

We sought the potential target cells of DSK signaling for regulating nociception. Since DSK receptor mutants were thermally hypersensitive consistently to *dsk* mutants (*Figure 1B, D and F*), neurons in the larval nociceptive circuit that express DSK receptors were promising candidates. To visualize the cells expressing DSK receptors, we generated T2A-GAL4 knock-ins in *CCKLR* genes. Both *CCKLR-17D1-T2A-GAL4* and *CCKLR-17D3-T2A-GAL4* were widely expressed in the larval CNS, predominantly in neuronal cells (*Figure 3A and B*; *Figure 3—figure supplement 1A–D*). By performing double-labeling experiments, we found that *CCKLR-17D1-T2A-GAL4* and *CCKLR-17D3-T2A-GAL4* are not expressed in the nociceptors at the periphery (*Figure 3C and D*; *Figure 3—figure supplement 1E* and F), or in nociceptive interneurons including Basin1-4 (*Ohyama et al., 2015*), A08n (*Kaneko et al., 2017*; *Hu et al., 2017*), and DnB neurons (*Burgos et al., 2018*) in the larval VNC (*Figure 3—figure supplement 1G–L*). However, we found that they are expressed in Goro neurons (*Figure 3E and F*), which are the fourth-order nociceptive interneurons located in the larval VNC (*Ohyama et al., 2015*).

To test whether the DSK receptors in Goro neurons are functionally important for regulating nociception, we performed RNAi and rescue experiments using *R69E06-GAL4* that marks Goro neurons (*Ohyama et al., 2015*). RNAi knockdown of CCKLR-17D1, but not CCKLR-17D3, in *R69E06-GAL4*

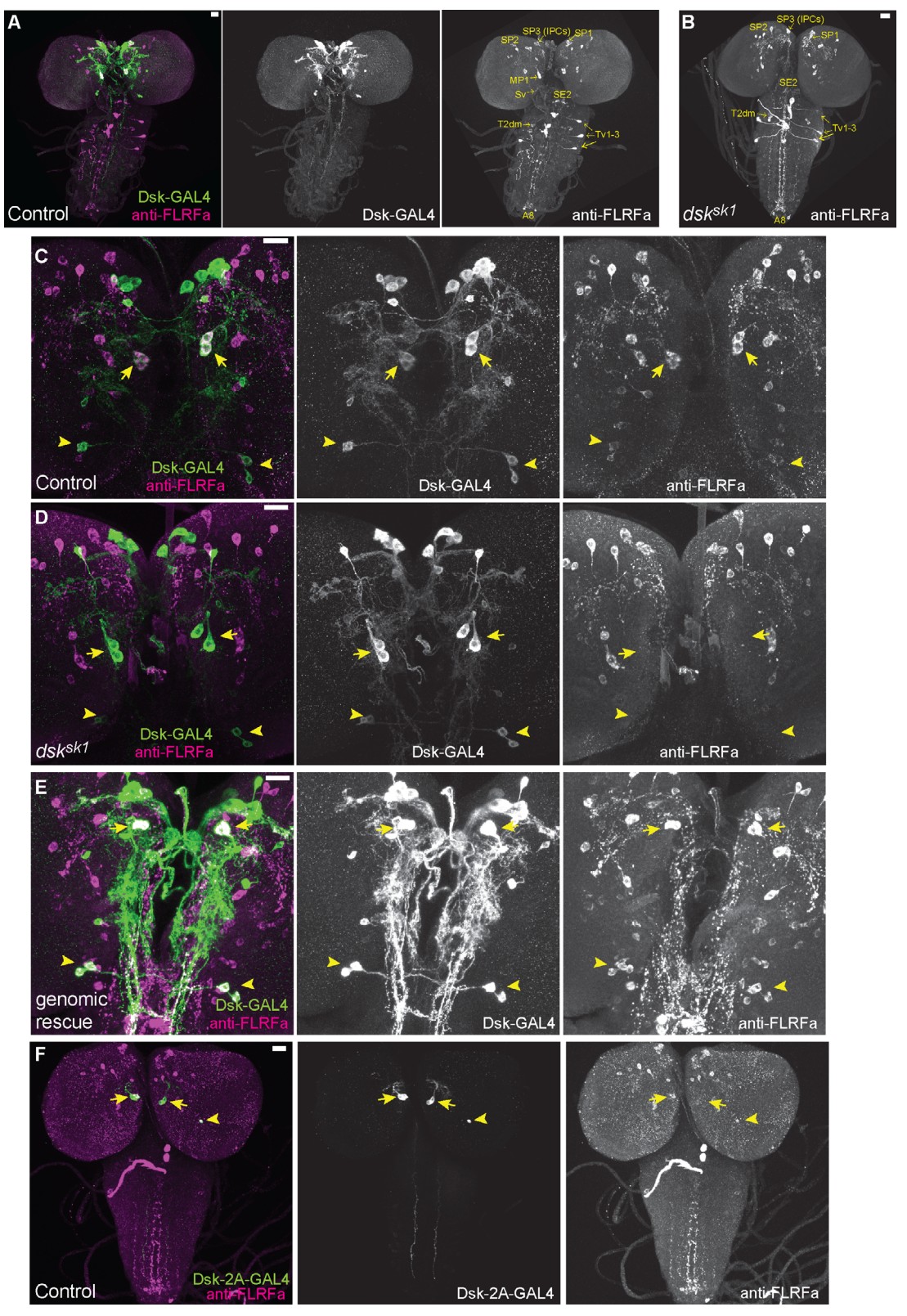

**Figure 2.** DSK is expressed in two groups of larval brain neurons. (**A**) Representative image of *DSK-GAL4* expression and anti-FLRFa staining in the wild-type larval CNS. (**B**) Representative image of anti-FLRFa staining in *dsk^{sk1}* larval CNS. (**C**) Representative images of *DSK-GAL4* expression and anti-FLRFa staining in the control larval brains. Arrows and arrowheads indicate MP1 and Sv neurons, respectively. Co-expressions of DSK-GAL4 and anti-FLRFa were observed in MP1 and Sv in all of the examined samples (n=27/27), but in IPCs only in 7% (n=2/27). (**D**) Representative images of *DSK-*

*Figure 2 continued on next page*

*Figure 2 continued*

GAL4 expression and anti-FLRFa staining in the *dsk^sk1^* larval brains. Arrows and arrowheads indicate MP1 and Sv neurons, respectively. (**E**) Representative images of *DSK-GAL4* expression and anti-FLRFa staining in the larval brains of the genomic rescue genotype (*Dp(3;1)2-2/Y; +/lexA-rCD2::RFP UAS-mCD8::GFP; dsk^sk1^ DSK-GAL4/dsk^sk1^*). Arrows and arrowheads indicate MP1 and Sv neurons, respectively. (**F**) An example image showing the expression of *DSK-2A-GAL4*, a 2A-GAL4 knock-in line of the *dsk* gene, more faithfully recapitulating its endogenous expression. Arrows and arrowheads indicate MP1 and Sv neurons, respectively. MP1 neurons were labeled in 100% of *DSK-2AGAL4* samples (n=41/41), while a single IPC in 51.2% (n=21/41), multiple IPCs in 24.4% (n=10/41), and Sv neurons in 17.1% (n=7/41) of examined samples. All scale bars represent 20 μm.

The online version of this article includes the following figure supplement(s) for figure 2:

**Figure supplement 1.** DSK expressions in the other brain neurons and peripheral sensory neurons.

neurons induced thermal hypersensitivity (*Figure 4A* and *Figure 4—figure supplement 1A*). *R69E06-GAL4* is also expressed in multiple neurons in the larval brain other than Goro neurons in the VNC (*Ohyama et al., 2015*). However, the thermal hypersensitivity was not observed when the expression of CCKLR-17D1 RNAi was excluded from Goro neurons by *tsh-GAL80*, pointing to the requirement of CCKLR-17D1 in Goro neurons (*Figure 4—figure supplement 1B–D*). Consistent with these RNAi results, expressing wild-type CCKLR-17D1, but not CCKLR-17D3, with *R69E06-GAL4* rescued the hypersensitivity of the respective mutants (*Figure 4B and C*). Furthermore, the morphology of Goro neurons was not affected by CCKLR-17D1 knockdown (*Figure 4—figure supplement 1E*). Therefore, we conclude that CCKLR-17D1 functions in Goro neurons to negatively regulate thermal nociception, but the function of CCKLR-17D3 in nociceptive regulations resides elsewhere.

Activation of Goro neurons elicits nocifensive rolling in larvae (*Ohyama et al., 2015*). Hence, if CCKLR-17D1 functions in Goro neurons to negatively regulate nociception, the lack of CCKLR-17D1 should sensitize these neurons to noxious heat. We directly addressed this hypothesis by using a Ca$^{2+}$ imaging technique we had previously developed (*Honjo and Tracey, 2018*; *Burgos et al., 2018*), whereby we monitored GCaMP6m signals in Goro neurons while locally applying thermal ramp stimuli to the larval body wall. In our thermal ramp stimulation protocol, the temperature around the larval CNS hardly reached 28 °C during the heat ramp stimulations (27.0 ± 0.2 °C at the peak, n=12), suggesting that the larval nociceptive system in the CNS that functions as an internal sensor for fast temperature increase (*Luo et al., 2017*) was unlikely to be activated.

Goro neurons in *CCKLR-17D1^Δ1^* mutants showed a significantly steeper GCaMP6m signal increase from 40 to 50 °C in comparison with the wild-type controls (*Figure 5A and B*, *Videos 1 and 2*). Since the baseline fluorescence levels of GCaMP6m were not significantly different in the wild-type and *CCKLR-17D1^Δ1^* mutants (*Figure 5C*), these data demonstrate that Goro neurons in *CCKLR-17D1^Δ1^* mutants are specifically sensitized to a noxious range of heat. Suppressing CCKLR-17D1 by RNAi in Goro neurons also induced significantly sensitized responses of Goro to noxious temperatures of 44–49 °C (*Figure 5D–F*, *Videos 3 and 4*). In contrast, Goro neurons in *CCKLR-17D3^Δ1^* mutants exhibited GCaMP6m signals that were mildly elevated but largely parallel compared with that of the controls (*Figure 5—figure supplement 1*, *Videos 5 and 6*), providing further evidence for the major functioning of CCKLR-17D3 in nociceptive regulation outside Goro neurons. Considered together, these data demonstrate that CCKLR-17D1 functions to negatively regulate the activity of Goro neurons, thereby attenuating behavioral nociceptive responses.

## Neuronal projections and thermal responsiveness of DSK-expressing neurons

If CCKLR-17D1 is involved in regulating the activity of Goro neurons, how can DSK be conveyed from the brain to the VNC? We noticed that some of the *DSK-GAL4* positive brain neurons sent descending neural processes to the VNC (*Figure 6A*), which were all anti-FLRFa positive (*Figure 6B*). Furthermore, the anti-FLRFa signals in the descending projections were completely absent in the *dsk* mutants (*Figure 6C*), suggesting that these descending projections likely originated from the DSK-expressing brain cells, namely MP1 and/or Sv. In analyzing *DSK-2A-GAL4*, we found that MP1 neurons in fact sent the descending projections to the VNC (*Figure 6D*). To further understand the projection patterns of MP1 and Sv neurons in detail, we performed single-cell labeling using an FLP-out technique and revealed that MP1 neurons sent descending projections contralaterally to the VNC (*Figure 6E*), while Sv neurons projected within the brain (*Figure 6F*). In comparison with the longitudinal processes to the VNC, MP1 neural processes in the brain possessed few anti-FLRFa positive puncta, which

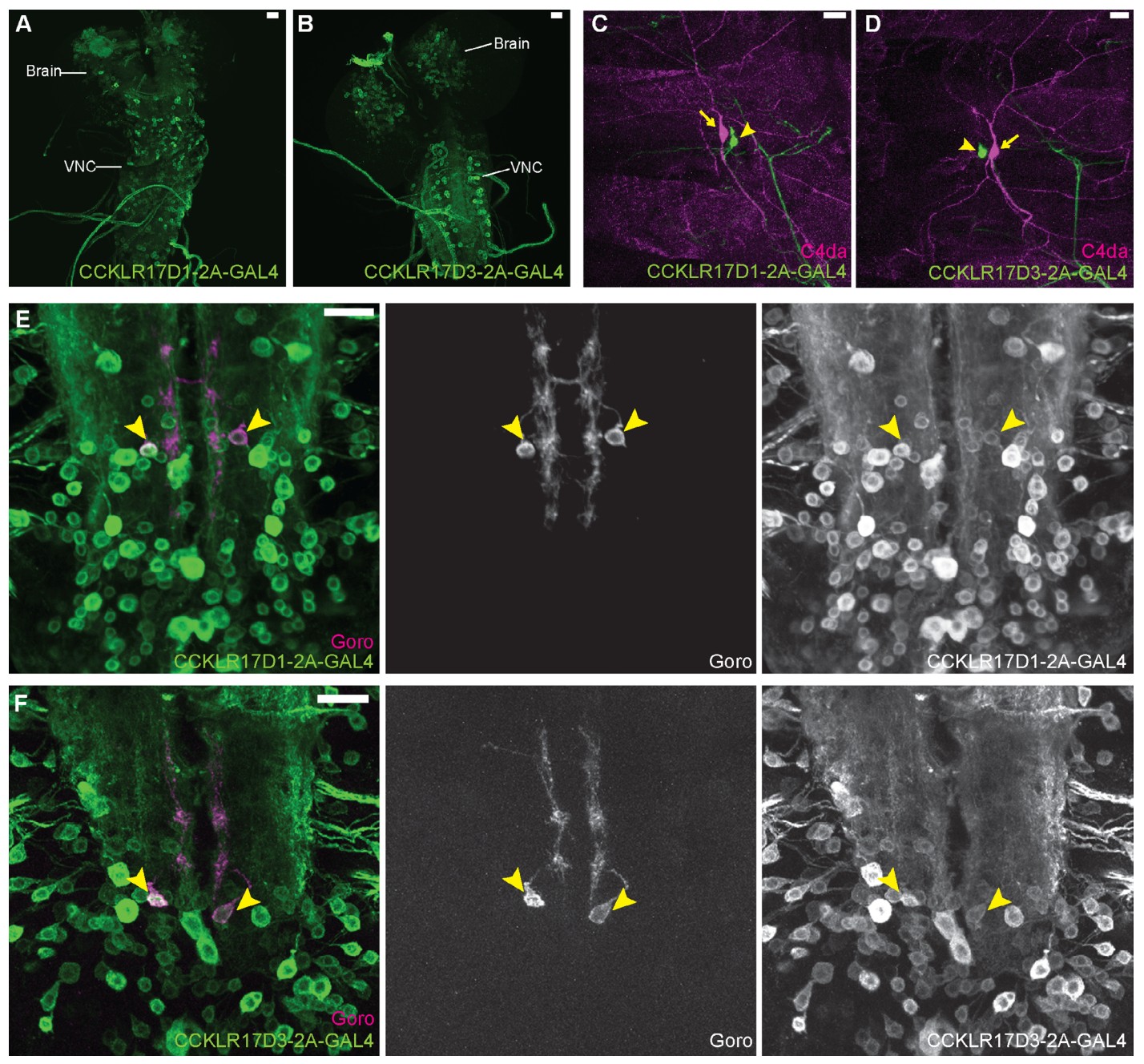

**Figure 3.** DSK receptors are expressed in Goro neurons in the larval VNC. (**A and B**) Representative images of *CCKLR-17D1-T2A-GAL4* (**A**) and *CCKLR-17D3-T2A-GAL4* (**B**) expressions in the larval CNS. (**C and D**) Representative images showing double-labeling of *CCKLR-17D1-T2A-GAL4* (**C**) and *CCKLR-17D3-T2A-GAL4* (**D**) with C4da nociceptors (*R38A10-lexA*). Arrows and arrowheads indicate cell bodies of C4da nociceptors and es cells, respectively. (**E and F**) Representative images showing double-labeling of *CCKLR-17D1-T2A-GAL4* (**E**) and *CCKLR-17D3-T2A-GAL4* (**F**) with Goro neurons (arrowheads, *R69E06-lexA*). Expression patterns were confirmed in multiple samples (n=7 and 5). All scale bars represent 20 μm.

The online version of this article includes the following figure supplement(s) for figure 3:

**Figure supplement 1.** Expression patterns of DSK receptors in larval glial cells, peripheral tissue, and nociceptive interneurons.

represent DSK in these neurons (*Figure 6G*). Furthermore, when the somatodendritic marker *UAS-Denmark* (*Nicolaï et al., 2010*) and the synaptic vesicle marker *UAS-syt::eGFP* were expressed in MP1 neurons, Denmark preferentially localized in the neural processes within the brain while syt::eGFP strongly accumulated in the processes descending to the VNC (*Figure 6H*). Collectively, these data demonstrate that MP1 neurons project DSK-positive descending axons to the VNC from the brain.

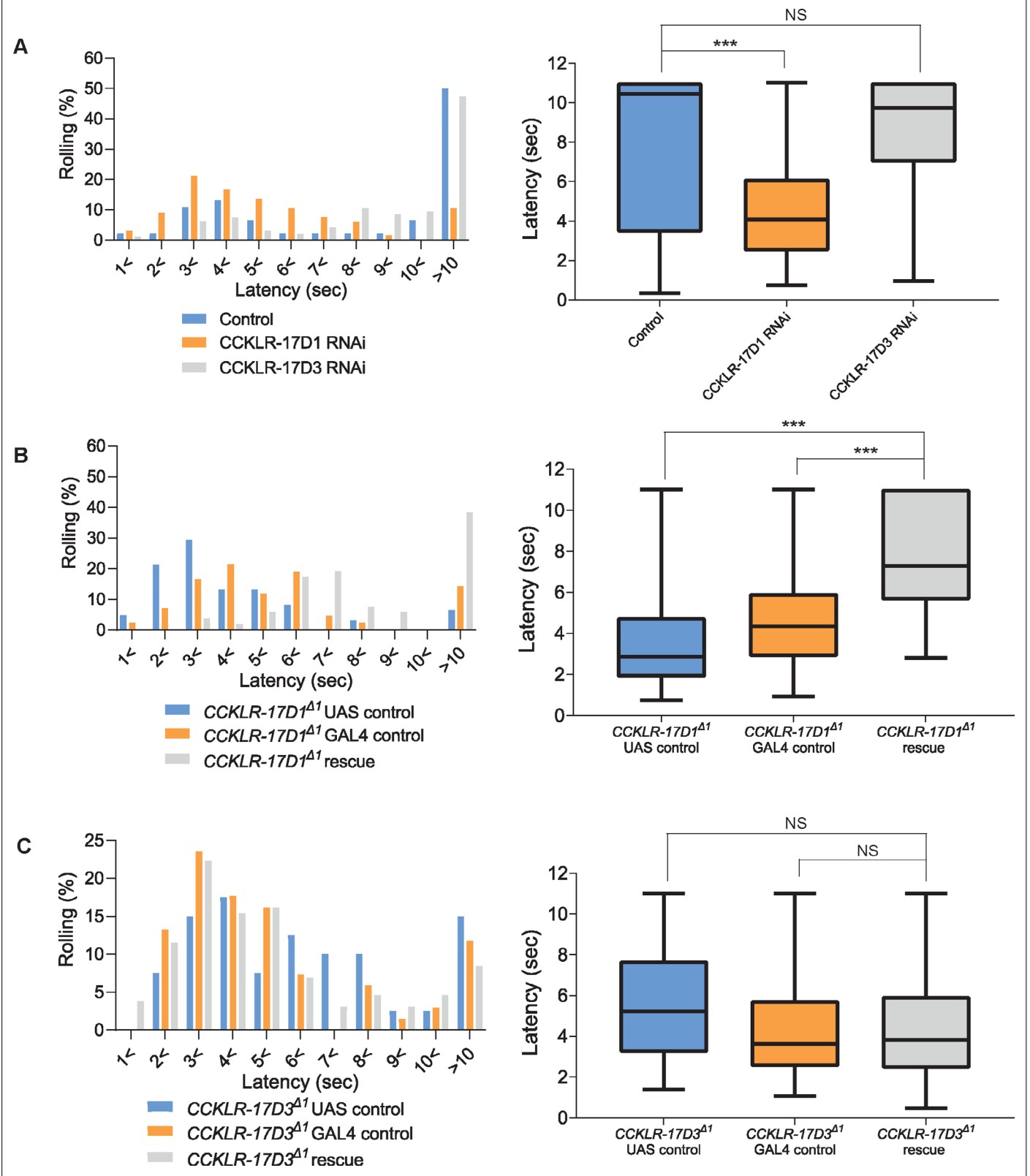

**Figure 4.** CCKLR-17D1 in Goro neurons is necessary and sufficient for normal thermal nociception. (**A**) RNAi of CCKLR-17D1 and CCKLR-17D3 using a Goro-GAL4 line *R69E06-GAL4*. Expressing CCKLR-17D1 RNAi (*R69E06-GAL4* x *yv; JF02644*, n=66) but not CCKLR-17D3 RNAi (*R69E06-GAL4* x *yv; JF02968*, n=95) with *R69E06-GAL4* caused significantly shorter latencies to 42 °C than controls (*R69E06-GAL4* x *yv; attp2*, n=46). (Left) Histograms (Right) Box plots of latencies. *** p<0.001, NS (non-significant) p>0.05 Steel's test. (**B**) Rescue of CCKLR-17D1 with *R69E06-GAL4*. The shortened latencies of

*Figure 4 continued on next page*

*Figure 4 continued*

*CCKLR-17D1*$^{\Delta 1}$ mutants observed in UAS controls (*CCKLR-17D1*$^{\Delta 1}$; *UAS-CCKLR-17D1/+*, n=61) or GAL4 controls (*CCKLR-17D1*$^{\Delta 1}$; *R69E06-GAL4/+*, n=42) were restored in the rescue genotype (*CCKLR-17D1*$^{\Delta 1}$; *UAS-CCKLR-17D1/+*; *R69E06-GAL4/+*, n=52). (Left) Histograms (Right) Box plots of latencies. *** p<0.001 Steel's test. (**C**) Rescue of CCKLR-17D3 with *R69E06-GAL4*. The shortened latencies of *CCKLR-17D3*$^{\Delta 1}$ mutants were unaltered in the rescue genotype (*CCKLR-17D3*$^{\Delta 1}$; *UAS-CCKLR-17D3/+*; *R69E06-GAL4/+*, n=130) compared with UAS controls (*CCKLR-17D3*$^{\Delta 1}$; *UAS-CCKLR-17D3/+*, n=40) or GAL4 controls (*CCKLR-17D3*$^{\Delta 1}$; *R69E06-GAL4/+*, n=68). (Left) Histograms (Right) Box plots of latencies. NS (non-significant) p>0.05 Steel's test. All box plots show median (middle line) and 25th to 75th percentiles with whiskers indicating the smallest to the largest data points.

The online version of this article includes the following source data and figure supplement(s) for figure 4:

**Source data 1.** Source data used to generate the summary data and graphs shown in *Figure 4* and *Figure 4—figure supplement 1*.

**Figure supplement 1.** CCKLR-17D1 RNAi in Goro neurons induces thermal hypersensitivity in a GAL4-dependent manner, without affecting the morphology.

Next, we investigated whether some of these DSK-expressing neurons are responsive to noxious thermal stimuli. In order to examine the responsiveness of multiple neurons in intact larvae, we took the snapshot approach using CaMPARI, a genetically encoded $Ca^{2+}$ sensor whose fluorescence irreversibly changes from the green to the red region of the spectrum in response to irradiation with 405 nm UV light in a $Ca^{2+}$ dependent manner, and is thus useful to monitor the activities of multiple neurons simultaneously in intact animals (*Fosque et al., 2015*). When the larvae expressing CaMPARI2 (an improved version of CaMPARI *Moeyaert et al., 2018*) in Goro neurons were stimulated by local thermal ramp stimuli under a 405 nm UV light, a significantly increased CaMPARI2 photoconversion was detected compared to the control group that had only been irradiated with 405 nm UV (*Figure 7A and B*), confirming that by using CaMPARI2 in our experimental setup we managed to successfully capture the increased activity of Goro neurons in response to noxious heat in intact larvae. We then tested the responsiveness of *DSK-2A-GAL4* neurons using the same stimulation and photoconversion protocol and found that none of the MP1 or Sv neurons, nor IPCs showed increased CaMPARI2 photoconversion in response to local thermal ramp stimuli (*Figure 7A and C–E*). Interestingly, MP1 neurons exhibited high CaMPARI2 photoconversion regardless of the presence of local heat ramp stimuli, which was even comparable to that observed in Goro neurons activated by noxious heat stimuli (*Figure 7A and C*). These data suggest that none of the *DSK-2A-GAL4* neurons are clearly responsive to noxious heat, but the markedly high CaMPARI2 photoconversion observed in MP1 neurons raises the possibility that these neurons may be active regardless of the presence of noxious heat stimuli.

## Connectivity between DSKergic and Goro neurons

To gain more insights into the connectivity between DSK-expressing brain neurons and Goro neurons, we further examined the anatomical relationship between MP1 axons and Goro neurons in the VNC. Performing double-labeling experiments, we found that MP1 axons with anti-FLRFa signals (representing DSK in MP1 axons) partially overlapped with Goro neurites in the thoracic segments of the larval VNC (*Figure 8A*). GFP reconstitution was reproducibly detected between MP1 and Goro neurons using the CD4-GRASP (GFP Reconstitution Across Synaptic Partners) system (*Feinberg et al., 2008*; *Gordon and Scott, 2009*), thus confirming that MP1 axons and Goro neurons are indeed in close proximity (*Figure 8B*). Since the CD4-GRASP is known to detect general cell-cell contacts (*Roy et al., 2014*; *González-Méndez et al., 2017*), we further investigated whether MP1 and Goro neurons are synaptically connected by using the nSyb-GRASP technique, which specifically enables synapse detection by localizing one of the split-GFP fragments in presynaptic sites (*Macpherson et al., 2015*). However, no signals of reconstituted GFP were detected between MP1 and Goro with nSyb-GRASP, although we found proximate localizations of nSyb-GFP$_{1-10}$ in the MP1 axons with the Goro neurites expressing CD4::GFP$_{11}$ (*Figure 8C*). We also employed the *trans*-Tango system (*Talay et al., 2017*) as another tool to detect synaptic connectivity. However, when the *trans*-Tango ligand was expressed by *DSK-2A-GAL4*, synaptic connectivity between MP1 and Goro was not indicated since the postsynaptic activation of *trans*-Tango signals was not observed in Goro neurons, although some *trans*-Tango-positive neurons were observed around Goro neurons (*Figure 8D*).

Although we could not find positive evidence for synaptic connectivity between MP1 axons and Goro neurons, we further tested the hypothesis that DSK-expressing neurons in the brain may have functional connectivity to Goro neurons. To achieve this, we first attempted artificial activation experiments of *DSK-2A-GAL4* neurons using *UAS-dTRPA1* to determine whether activating DSK-expressing

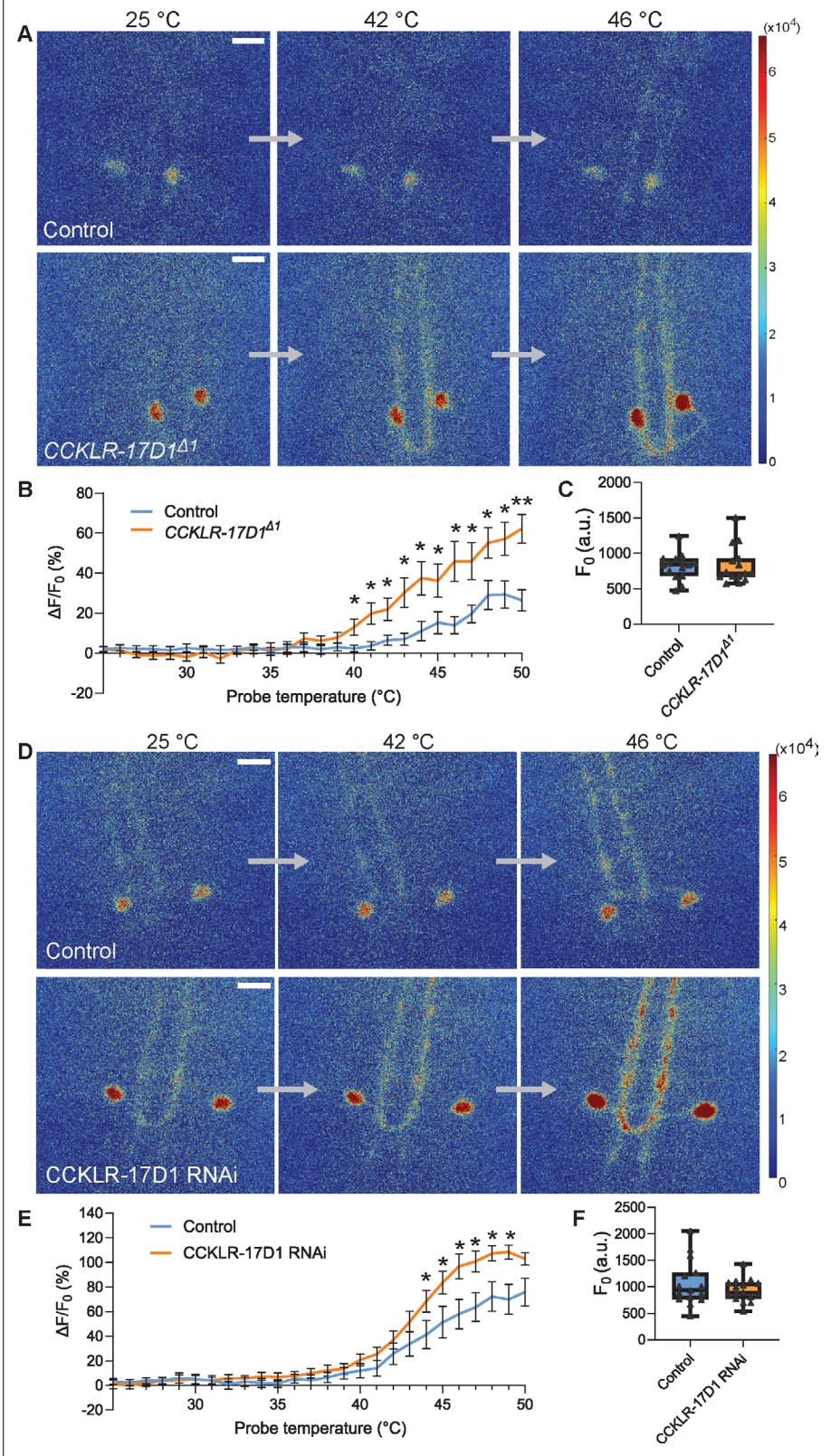

**Figure 5.** Goro neurons lacking CCKLR-17D1 show sensitized responses to noxious heat. (**A**) Representative still images showing thermal activation of Goro neurons in the controls (top, *yw/Y; R69E06-GAL4 UAS-GCaMP6m/+*) and *CCKLR-17D1^Δ1* mutants (bottom, *CCKLR-17D1^Δ1^/Y; R69E06-GAL4 UAS-GCaMP6m/+*). See also *Videos 1 and 2*. (**B**) Average percent increase of GCaMP6m fluorescence intensity relative to the baseline (ΔF/F₀) during

*Figure 5 continued on next page*

*Figure 5 continued*

heat ramp stimulations in controls (n=15) and *CCKLR-17D1*$^{\Delta 1}$ (n=15). * p<0.05, ** p<0.01 Mann-Whitney's U-test. (**C**) Basal GCaMP6m signal levels (**F**$_0$) in controls (n=15) and *CCKLR-17D1*$^{\Delta 1}$ mutants (n=15). p>0.6 Mann-Whitney's U-test. (**D**) Representative stills showing thermal activation of the controls (top, *R69E06-GAL4 UAS-GCaMP6m* x *yv; attp2*) and Goro neurons expressing CCKLR-17D1 RNAi (bottom, *R69E06-GAL4 UAS-GCaMP6m* x *yv; JF02644*). See also *Videos 3 and 4*. (**E**) Average percent increase of GCaMP6m fluorescence intensity relative to the baseline (ΔF/F$_0$) during heat ramp stimulations in controls (n=15) and CCKLR-17D1 RNAi (n=16). * p<0.05 Mann-Whitney's U-test. (**F**) Basal GCaMP6m levels (**F**$_0$) in controls (n=15) and CCKLR-17D1 RNAi (n=16). p>0.6 Mann–Whitney's U-test. Error bars represent standard error. All box plots show median (middle line) and 25th to 75th percentiles with whiskers indicating the smallest to the largest data points. All scale bars represent 20 μm.

The online version of this article includes the following source data and figure supplement(s) for figure 5:

**Source data 1.** Source data used to generate the summary data and graphs shown in *Figure 5* and *Figure 5—figure supplement 1*.

**Figure supplement 1.** Thermal responsiveness of Goro neurons in *CCKLR-17D3*$^{\Delta 1}$.

neurons actually attenuates larval nociception. dTRPA1 is a cation channel gated by warm temperature (>29 °C) that has been used as a tool to artificially activate neurons of interest in *Drosophila* (*Hamada et al., 2008*). When the larvae expressing dTRPA1 in C4da nociceptors with *ppk1.9-GAL4* were placed in a chamber at 35 °C, dTRPA1-induced thermogenetic activation of C4da nociceptors caused nocifensive rolling responses within two seconds in the majority of animals, as reported previously (*Zhong et al., 2012*; *Figure 9A and B*). In contrast, when the larvae expressing dTRPA1 simultaneously in *ppk1.9-GAL4* and *DSK-2A-GAL4* were placed in a chamber at 35 °C, the additional activation of *DSK-2A-GAL4* neurons to C4da nociceptors resulted in a markedly reduced percentage of larvae showing rolling responses within 2 s and in significantly lengthened latencies to exhibit a rolling response (*Figure 9A and B*). None of the animals expressing dTRPA1 with *DSK-2A-GAL4* alone showed a rolling response within 10 s (*Figure 9A*).

The above data suggest that activation of *DSK-2A-GAL4* neurons inhibits larval nociception. We then investigated whether activating *DSK-2A-GAL4* neurons could attenuate the activity of Goro neurons, by combining the artificial activation experiments described above with CaMPARI2 snapshot imaging of Goro neurons. Indeed, CaMPARI2 photoconversion in Goro neurons induced by the thermogenetic

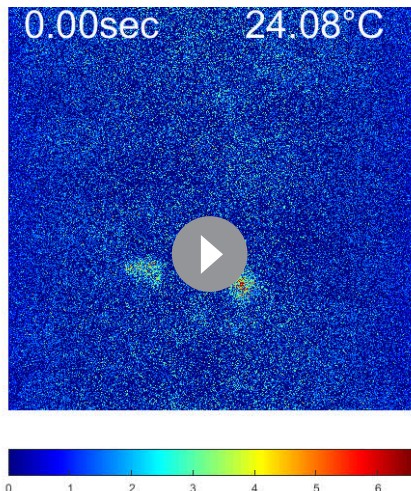

**Video 1.** Ca$^{2+}$ imaging of Goro neurons in a control animal used for the *CCKLR-17D1*$^{\Delta 1}$ experiments. See also *Figure 5*. The movie was generated from heat-mapped time-series projection images of GCaMP6m fluorescence using MATLAB. Time and probe temperature are indicated at the top left and top right corners, respectively.

https://elifesciences.org/articles/85760/figures#video1

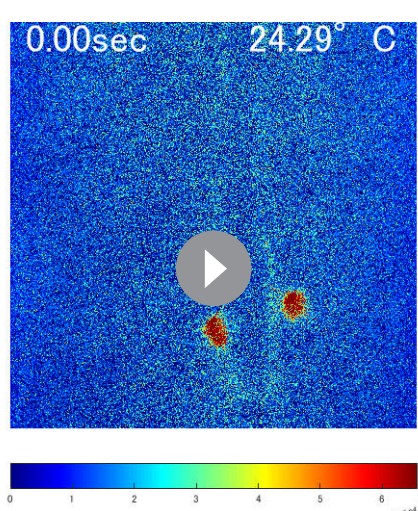

**Video 2.** Ca$^{2+}$ imaging of Goro neurons in a *CCKLR-17D1*$^{\Delta 1}$ mutant animal. See also *Figure 5*. The movie was generated from heat-mapped time-series projection images of GCaMP6m fluorescence using MATLAB. Time and probe temperature are indicated at the top left and top right corners, respectively.

https://elifesciences.org/articles/85760/figures#video2

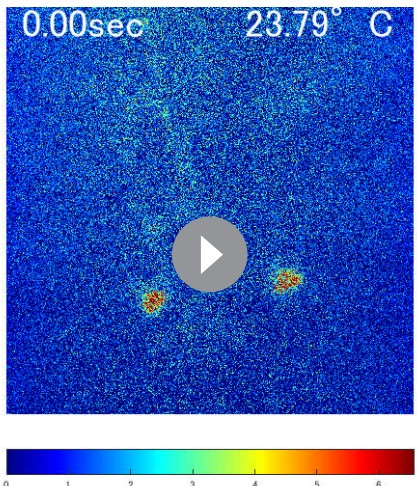

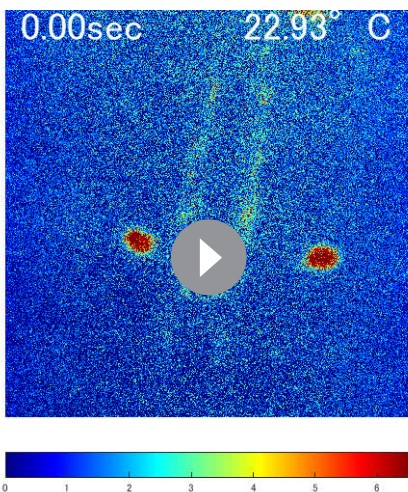

**Video 3.** Ca²⁺ imaging of Goro neurons in a control animal used for the CCKLR-17D1 RNAi experiments. See also *Figure 5*. The movie was generated from heat-mapped time-series projection images of GCaMP6m fluorescence using MATLAB. Time and probe temperature are indicated at the top left and top right corners, respectively.

https://elifesciences.org/articles/85760/figures#video3

**Video 4.** Ca²⁺ imaging of Goro neurons in a CCKLR-17D1 RNAi animal. See also *Figure 5*. The movie was generated from heat-mapped time-series projection images of GCaMP6m fluorescence using MATLAB. Time and probe temperature are indicated at the top left and top right corners, respectively.

https://elifesciences.org/articles/85760/figures#video4

activations of C4da nociceptors was significantly decreased by 35% with the simultaneous activation of *DSK-2A-GAL4* neurons (*Figure 9C and D*). Given that MP1 neurons were the only cells that were labeled by *DSK-2A-GAL4* with 100% reproducibility, while the expression of *DSK-2A-GAL4* in IPCs and Sv neurons was minor and stochastic (*Figure 2F*), the observed inhibition of behavioral nociception and Goro neurons caused by activating *DSK-2A-GAL4* neurons is mostly attributable to the activation

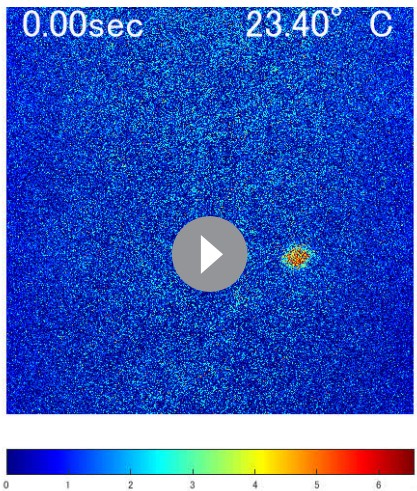

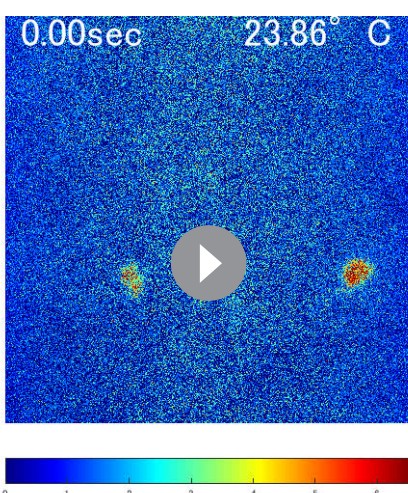

**Video 5.** Ca²⁺ imaging of Goro neurons in a control animal used for the *CCKLR-17D3^Δ1* experiments. See also *Figure 5—figure supplement 1*. The movie was generated from heat-mapped time-series projection images of GCaMP6m fluorescence using MATLAB. Time and probe temperature are indicated at the top left and top right corners, respectively.

https://elifesciences.org/articles/85760/figures#video5

**Video 6.** Ca²⁺ imaging of Goro neurons in a *CCKLR-17D3^Δ1* mutant animal. See also *Figure 5—figure supplement 1*. The movie was generated from heat-mapped time-series projection images of GCaMP6m fluorescence using MATLAB. Time and probe temperature are indicated at the top left and top right corners, respectively.

https://elifesciences.org/articles/85760/figures#video6

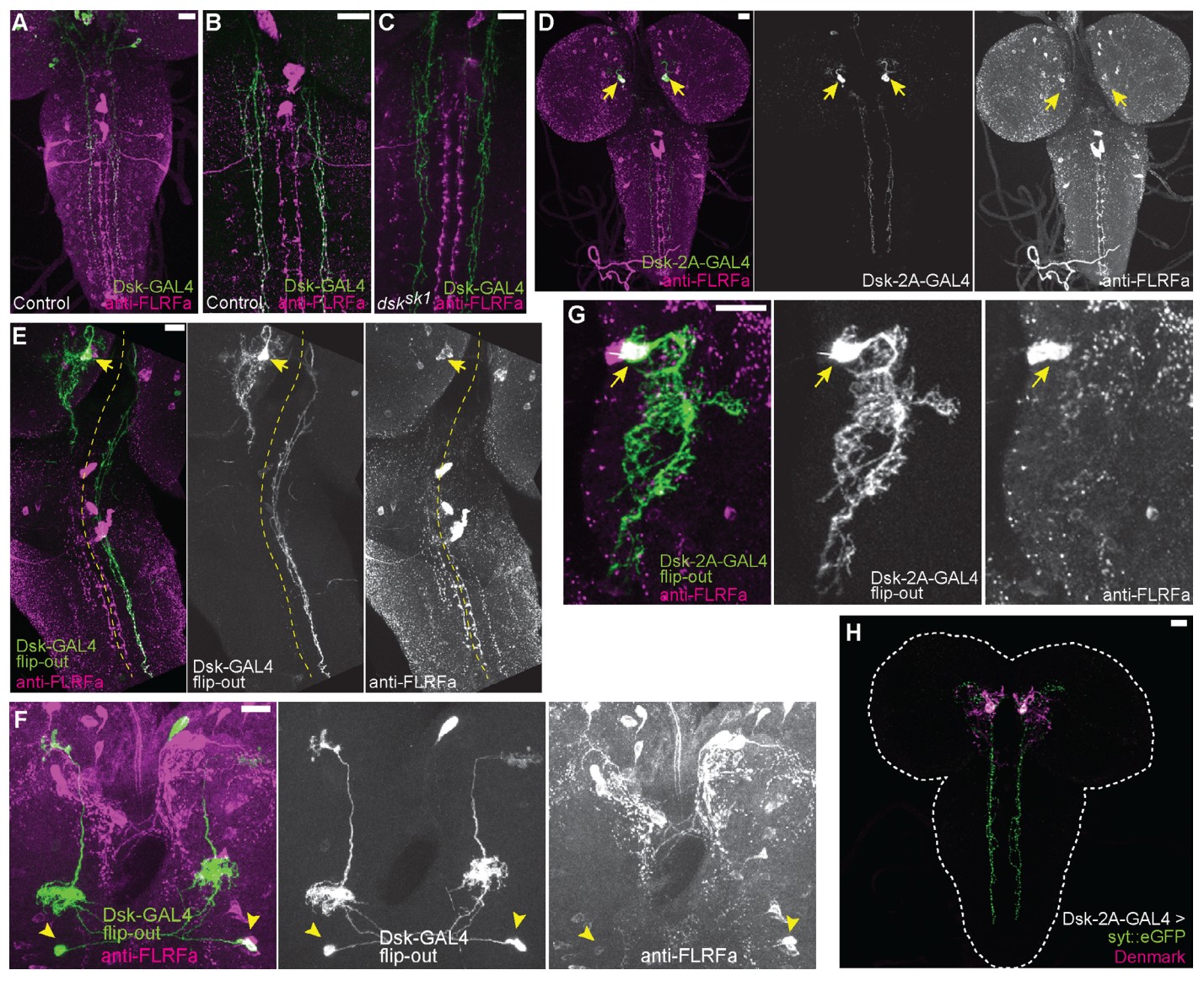

**Figure 6.** Projection patterns of DSK-expressing neurons in the larval CNS. (**A**) Representative image showing descending axons from *DSK-GAL4* positive brain neurons (green) to the larval VNC. (**B**) Representative image showing all descending projections labeled by *DSK-GAL4* (green) harbor punctate anti-FLRFa signals (magenta) in the wild-type. (**C**) Representative image showing the complete absence of punctate anti-FLRFa signals (magenta) in the descending projections labeled by *DSK-GAL4* (green) in the *dsk^{sk1}* mutants. (**D**) Image showing the descending projections of MP1 neurons (arrows) marked by *DSK-2A-GAL4*. (**E**) Image showing a single FLP-out clone of MP1 neuron (arrow) contralaterally sending descending axons to the larval VNC. The yellow dashed line indicates the midline. (**F**) Image showing two FLP-out clones of Sv neurons (arrowheads) sending ascending axons to both sides of the brain. (**G**) A projection image showing the MP1 neurites in the brain. Note that few anti-FLRFa-positve puncta are associated with MP1 neurites within the brain. Similar expression patterns were observed in multiple samples (n=3). The arrow indicates the anti-FLRFa-positve MP1 soma. (**H**) Representative image showing localizations of syt::eGFP (green) and Denmark (magenta) in MP1 neurons (*DSK-2A-GAL4* x *UAS-syt::eGFP UAS-Denmark*). Similar expression patterns were observed in multiple samples (n=7). All scale bars represent 20 μm.

of MP1 neurons. Thus, these data support the functional connectivity between DSKergic MP1 neurons in the brain and Goro neurons.

## Discussion

In this study, we have demonstrated that (1) DSK and its receptors CCKLR-17D1 and CCKLR-17D3 are involved in negatively regulating thermal nociception. (2) Two sets of brain neurons, MP1 and Sv, express DSK in the larval nervous system. (3) One of the DSK receptors (CCKLR-17D1) functions in

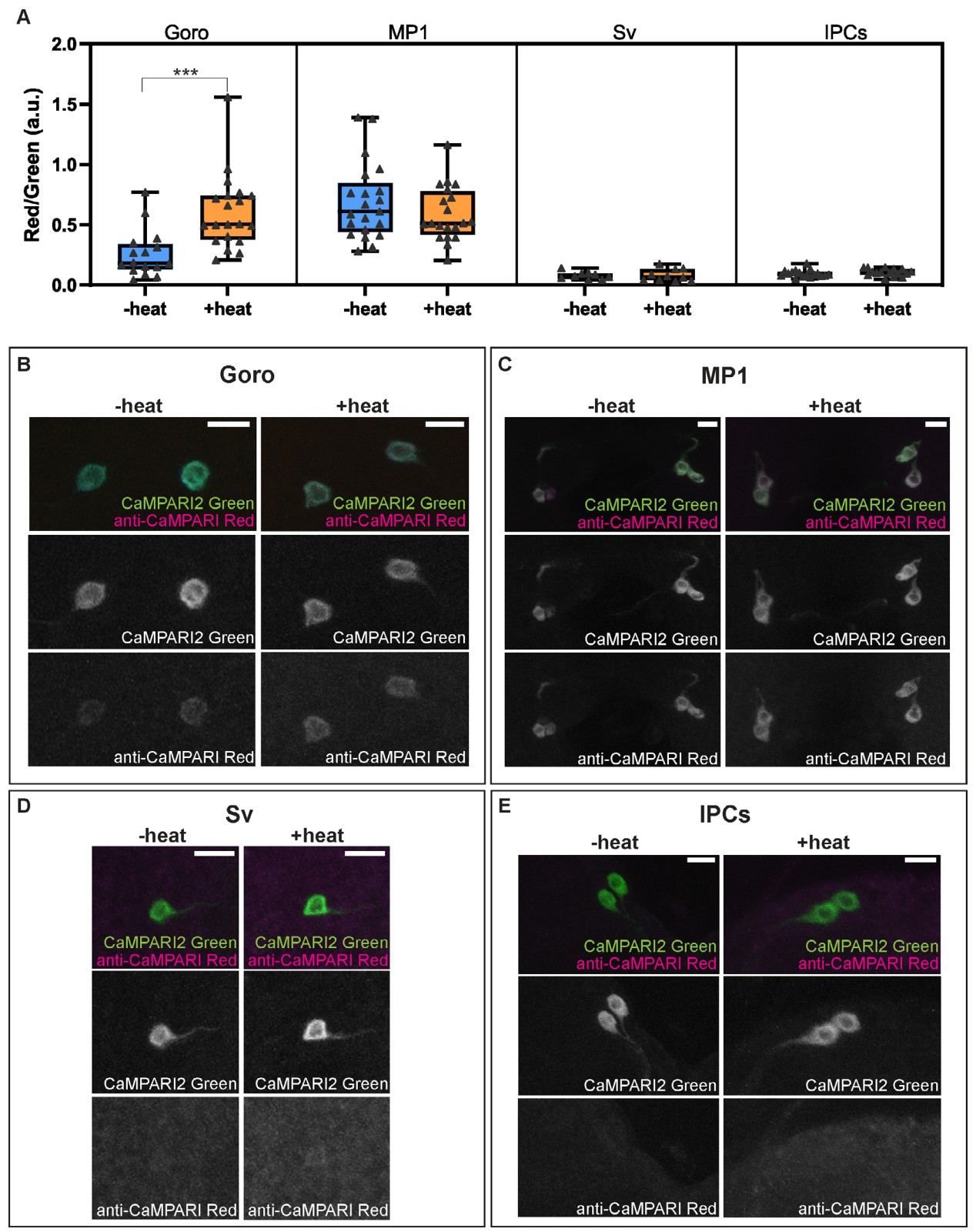

**Figure 7.** Activity and thermal responsiveness of DSK-expressing neurons. (**A**) Box plots of the quantified ratio of CaMPARI2 green signals and photoconverted CaMPARI2 red signals detected by anti-CaMPARI-red in Goro, MP1, Sv, and IPCs (Red/Green) with or without local heat ramp stimulation. Goro neurons showed a significantly higher Red/Green ratio of CaMPARI2 when local heat ramp stimulation was applied to the larvae (n=16 and 20, p<0.001, Mann-Whitney's U-test). MP1 neurons showed a high Red/Green ratio of CaMPARI2 both with and without local heat

*Figure 7 continued on next page*

*Figure 7 continued*

stimulation (n=21 and 20, p>0.3, Mann-Whitney's U-test). The Red/Green ratio of CaMPARI2 was low in Sv compared to MP1 neurons, and CaMPARI2 photoconversion was not increased by local heat stimulation (n=9 and 10, p>0.5, Mann-Whitney's U-test). IPCs also exhibited low CaMPARI2 photoconversion regardless of the presence of local heat ramp stimulation (n=17 and 16, p>0.25, Mann-Whitney's U-test). (**B–E**) Representative images showing CaMPARI2 green signals (green) and anti-CaMPARI-red signals (magenta) in Goro (**B**), MP1 (**C**), Sv (**D**), and IPCs (**E**). All box plots show median (middle line) and 25th to 75th percentiles, with whiskers indicating the smallest to the largest data points. Scale bars represent 10 μm.

The online version of this article includes the following source data for figure 7:

**Source data 1.** Source data used to generate the summary data and graphs shown in *Figure 7*.

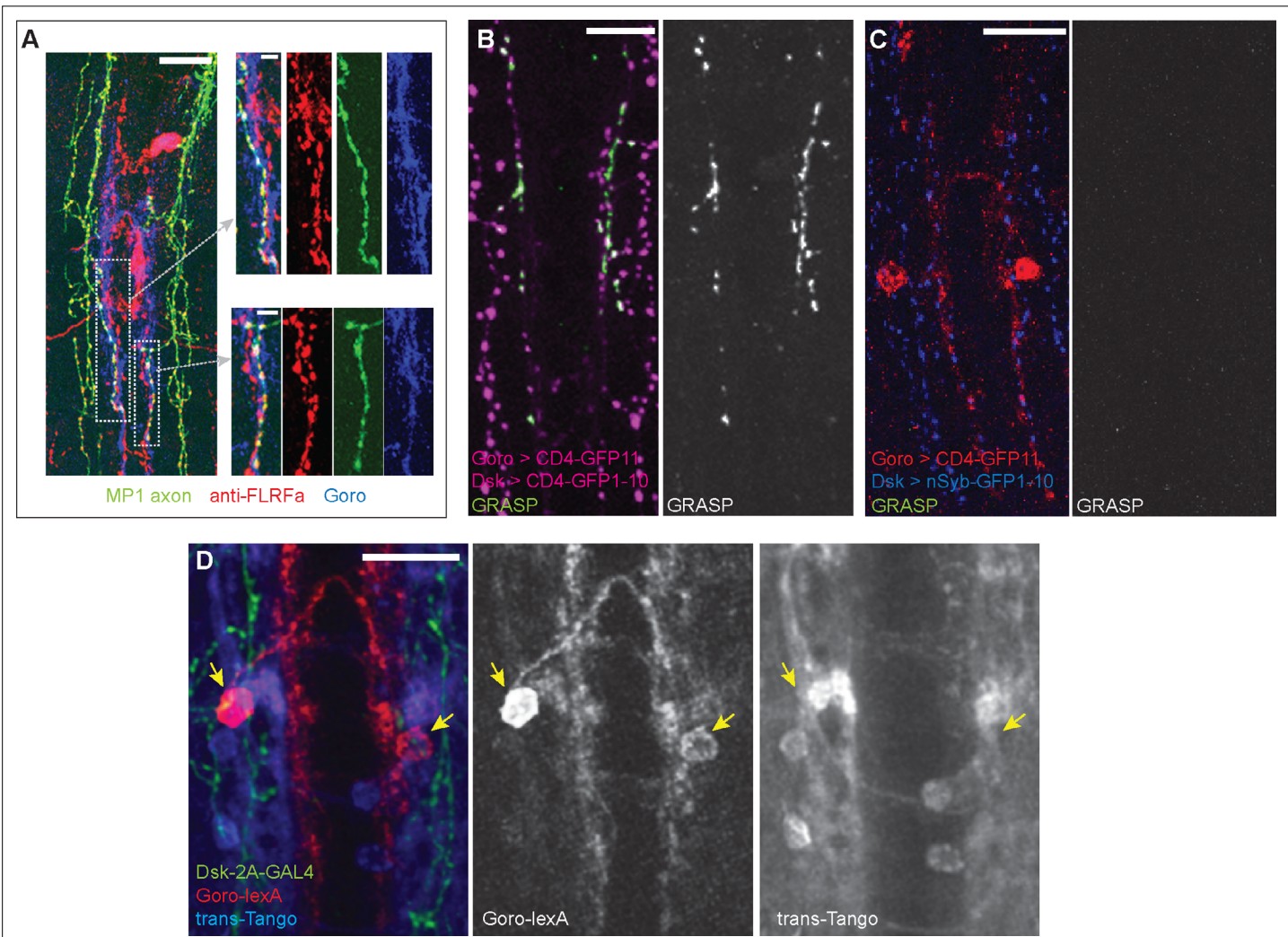

**Figure 8.** Synaptic connectivity between MP1 axons and Goro neurons is not detected. (**A**) Representative images of the larval VNC showing partially overlapping MP1 axons (green) and processes of Goro neurons (blue) that are associated with anti-FLRFa signals (red). MP1 axons partially overlapping Goro neurons were similarly observed in all examined samples (n=5). (**B**) Representative image showing CD4-GRASP experiments between MP1 axons and Goro neurons (magenta). Signals of reconstituted GFP (GRASP, green) were detected in all examined samples (n=5). (**C**) Representative image showing nSyb-GRASP experiments between MP1 axons (blue) and Goro neurons (red). Signals of reconstituted GFP (GRASP, green) were not detectable in any of the examined samples (n=5). (**D**) Representative image showing *trans*-Tango experiments using *DSK-2A-GAL4* (*R69E06-lexA, lexAop-rCD2::RFP UAS-mCD8::GFP; DSK-2A-GAL4 x UAS-myrGFP, QUAS-mtdTomato::3xHA; trans-Tango*). Transsynaptic activation of Tango signals (blue; mtdTomato::3xHA detected by anti-HA) was undetectable in Goro neurons (red; rCD2::RFP detected by anti-CD2). Green represents MP1 axons labeled by *DSK-2A-GAL4* (myrGFP and mCD8::GFP detected by anti-GFP). Signals of mtdTom::3xHA in Goro neurons were not detected in any of the examined samples (n=4). Arrows indicate Goro somata. All scale bars represent 20 μm, except for the insets in (**A**) (5 μm).

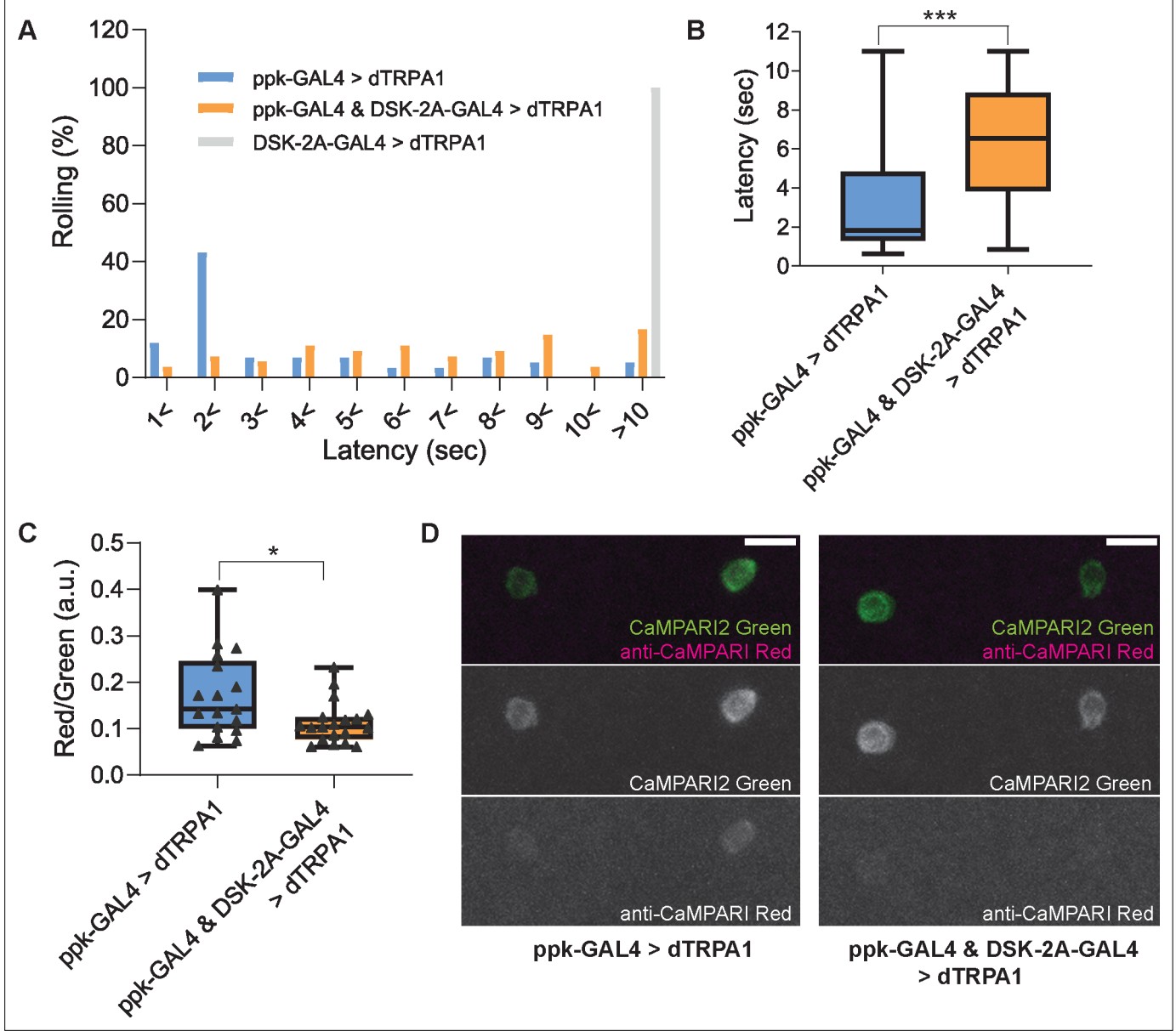

**Figure 9.** Thermogenetic activation of *DSK-2A-GAL4* neurons inhibits larval nociception and Goro neurons. (**A and B**) Thermogenetic activations of nociceptors and/or *DSK-2A-GAL4* neurons using *UAS-dTRPA1*. (**A**) Histogram representing the percentage of larvae exhibiting nociceptive rolling. Upon thermogenetic activations, 55% of larvae expressing dTRPA1 in nociceptors (ppk-GAL4 >dTRPA1; *ppk1.9-GAL4* x *UAS-dTRPA1*, n=58) exhibited nociceptive rolling within 2 s. In contrast, only 11% of larvae with simultaneous activations of nociceptors and *DSK-2A-GAL4* showed rolling within 2 s (ppk-GAL4 & DSK-2A-GAL4>dTRPA1; *ppk1.9-GAL4 DSK-2A-GAL4* x *UAS-dTRPA1*, n=54). Thermogenetic activations of *DSK-2A-GAL4* neurons alone (DSK-2A-GAL4>dTRPA1; *DSK-2A-GAL4* x *UAS-dTRPA1*, n=40) did not trigger nociceptive responses even after 10 s. (**B**) Box plots showing latencies. Thermogenetic activation of *DSK-2A-GAL4* simultaneously with nociceptors caused a significantly longer latency to rolling responses (Mann-Whitney U-test, *** $p<0.001$). (**C and D**) CaMPARI2 snapshot imaging of the activity of Goro neurons activated by thermogenetic stimulation of C4da nociceptors. (**C**) Box plots of the ratio of CaMPARI2 green signals and photoconverted CaMPARI2 red signals detected by anti-CaMPARI-red in Goro neurons (Red/Green). Compared to the larvae with thermogenetic activation of C4da nociceptors only (ppk-GAL4 >dTRPA1; *ppk1.9-GAL4, UAS-dTRPA1* x *R69E06-lexA; lexAop-CaMPARI2*; n=17), the larvae with simultaneous activation of C4da and DSK-2A-GAL4 neurons (ppk-GAL4 & DSK-2A-GAL4>dTRPA1; *ppk1.9-GAL4, UAS-dTRPA1; DSK-2A-GAL4* x *R69E06-lexA; lexAop-CaMPARI2*; n=19) showed significantly decreased CaMPARI2 photoconversion in Goro neurons (Mann-Whitney U-test, * $p<0.025$). (**D**) Representative images showing CaMPARI2 green signals (green) and anti-CaMPARI-red signals (magenta) in Goro neurons of animals with thermogenetic C4da activation (ppk-GAL4 >dTRPA1) and with simultaneous thermogenetic activation of C4da and DSK-2A-GAL4 neurons (ppk-GAL4 & DSK-2A-GAL4>dTRPA1). All box plots show median (middle line) and 25th to 75th percentiles with whiskers indicating the smallest to the largest data points. Scale bars represent 10 µm.

The online version of this article includes the following source data for figure 9:

**Source data 1.** Source data used to generate the summary data and graphs shown in *Figure 9*.

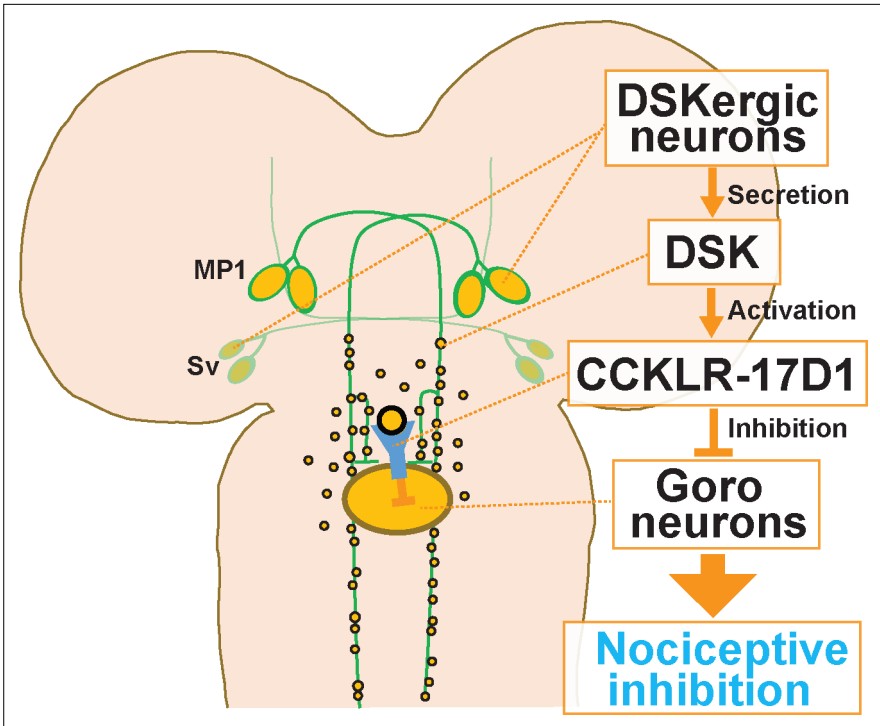

**Figure 10.** A schematic model of the DSKergic descending inhibitory pathway of nociception in larval *Drosophila*. DSK-expressing MP1 and/or Sv neurons in the brain secrete DSK peptides. DSK in the larval VNC activates the CCKLR-17D1 receptor expressed in Goro neurons, which subsequently inhibits the activity of Goro neurons, and ultimately larval nociceptive rolling responses.

Goro neurons in the VNC to negatively regulate thermal nociception, and (4) Thermogenetic activation of *DSK-2A-GAL4* neurons attenuates both larval nociception and the activity of Goro neurons. Based on these results, we propose that the DSKergic neurons regulating the activity of Goro neurons constitute a descending inhibitory pathway of nociception from the brain to the VNC in larval *Drosophila* (*Figure 10*). To our knowledge, our findings represent the first evidence of a descending mechanism modulating nociception from the brain in a non-mammalian species.

## DSKergic descending system as a physiological modulator of nociception

DSK has been implicated in multiple physiological and developmental processes in *Drosophila* (*Nässel and Williams, 2014*; *Wu et al., 2019*; *Wu et al., 2020*; *Mohammad et al., 2016*). Previous studies have shown that *dsk* and *CCKLR-17D1* mutants exhibit significant reductions of synaptic growth and excitability in larval neuromuscular junctions (NMJ) and larval locomotion under bright light (*Chen et al., 2012*; *Chen and Ganetzky, 2012*), suggesting the importance of the DSK/CCKLR-17D1 signaling pathway in the developmental processes of motoneurons. However, in this study, no major developmental defects were observed in Goro neurons with CCKLR-17D1 RNAi (*Figure 4—figure supplement 1E*). Furthermore, the simultaneous thermogenetic activations of *DSK-2A-GAL4* neurons and C4da nociceptors inhibited larval nociception and the activity of Goro neurons (*Figure 9*). Given the pronociceptive role of Goro neurons, their reduced synaptic or neuronal activity should cause nociceptive insensitivity or reduced $Ca^{2+}$ responses, which is contradicting to our observation that CCKLR-17D1 knockdown in Goro produced the thermal hypersensitivity and exaggerated $Ca^{2+}$ responses (*Figures 4 and 5*). Thus, these data consistently support a physiological role for DSKergic descending signals in the modulation of the activity of Goro neurons rather than a developmental role, highlighting the functional differences of the DSK/CCKLR-17D1 pathway between the NMJ and the nociceptive system in the CNS.

## DSK in larval IPCs dispensable for nociceptive modulation

A previous study has reported the expression of DSK in a population of larval IPCs and its functions in responding to starving conditions (*Söderberg et al., 2012*). However, we observed that anti-FLRFa signals in IPCs persisted in *dsk* null mutants and that anti-FLRFa signals in IPCs hardly overlap with either *DSK-GAL4* or *DSK-2A-GAL4* expressions (*Figure 2* and *Figure 2—figure supplement 1*). Although we used a different antibody from that in the study mentioned above, the number of IPCs visualized by anti-FLRFa was comparable to that of cells visualized by anti-DSK (*Figure 2—figure supplement 1*; *Söderberg et al., 2012*). Regarding the functions of IPCs in nociception, Im et al. showed that silencing *dilp2-GAL4* positive IPCs has no significant effect on the baseline nociceptive responses (*Im et al., 2018*). In our CaMPARI2 imaging, IPCs marked by *DSK-2A-GAL4* showed very low baseline activity and no responsiveness to noxious heat (*Figure 7*). Thus, the current and previous studies strongly suggest that IPCs are likely irrelevant for DSK-mediated nociceptive regulation, at least under normal conditions, although it is still possible for DSK to be expressed stochastically in a limited population of IPCs.

## Functional mechanisms of DSKergic nociceptive inhibitory system

Although our data suggest the existence of a DSKergic descending inhibitory system from the brain to the Goro neurons in the VNC, there are multiple remaining questions for future studies on the mechanisms by which DSK-expressing neurons in the brain regulate Goro neurons: First, whether both MP1 and Sv neurons are involved in nociceptive inhibition remains unclear. Our findings consistently point to MP1 neurons over Sv neurons as a major source of DSK regulating Goro neurons. However, given the potential of DSK for distant actions through diffusion, the possibility of Sv neurons communicating with Goro neurons cannot be eliminated. Further analysis using finer genetic/neuronal tools will be necessary to tease the functions of MP1 and Sv neurons apart and understand the functional mechanism of the DSKergic system in the regulation of larval nociception.

Second, how this DSKergic system is activated to regulate nociception in vivo needs further investigation. We observed very high CaMPARI2 photoconversion in MP1 neurons without noxious heat stimuli, which did not further increase in response to noxious heat application (*Figure 7*). These data raise the hypothesis that MP1 neurons may function as a tonically active brake for nociceptive circuits, rather than a simple negative feedback system that is activated by nociceptive stimuli as input. The tonically active model of MP1 function is also consistent with our data showing that *dsk* and *CCKLR-17D1* mutations, as well as CCKLR-17D1 RNAi in Goro neurons, all caused thermal hypersensitivity under normal conditions (*Figures 1 and 4*). Up to this point, no external or internal signals have been directly shown to be the input into larval DSKergic neurons. However, since DSK has been implicated in the regulation of several physiological processes such as hunger and stress, sensory or molecular signals that are involved in these physiological processes have the potential to serve as upstream cues that activate DSKergic neurons. Alternatively, it is also possible that MP1 neurons may have spontaneous activity. Further physiological studies on the activity and responsiveness of DSKergic neurons would be required to elucidate how the DSKergic system functions to regulate nociception in larvae under normal conditions.

Third, the transmission mechanisms of DSK from the brain to Goro neurons need further clarification. Our nSyb-GRASP and *trans*-Tango experiments consistently showed negative results, indicating no synaptic connectivity between MP1 descending axons and Goro neurons (*Figure 8*). Thus, the interaction between DSK from DSKergic neurons in the brain and Goro neurons in the VNC may be mediated non-synaptically through volume transmission, as described in many neuropeptidergic systems (*van den Pol, 2012*). Many of the CCKLR-expressing neurons in the VNC are distantly located from the descending MP1 axons (*Figures 3 and 6*). Since it is unlikely that all these CCKLR-expressing neurons are synaptically connected to MP1 axons, neuronal communications through volume transmission can be fairly assumed for DSKergic systems in the larval VNC. More detailed analyses at an electron microscopic level of the circuitry connectivity between MP1 and Goro neurons and of the localization of DSK-containing vesicles in DSKergic neurons as well as DSK receptors in Goro neurons would be necessary to further clarify the transmission mechanisms of DSK from the brain to Goro neurons.

## Multiple molecular pathways for DSK signaling in the regulation of nociception

The data presented in this study also suggest that DSK signaling could regulate larval nociception through multiple pathways other than the DSK/CCKLR-17D1 system. For example, while *CCKLR-17D3* mutants exhibited as severe thermal hypersensitivity as *CCKLR-17D1* mutants (*Figure 1D and F*), our RNAi and rescue experiments failed to locate the function of CCKLR-17D3 to Goro neurons (*Figure 4A and C*). Our GCaMP Ca$^{2+}$ imaging also revealed that Goro neurons lacking CCKLR-17D3 showed modestly sensitized responses to noxious heat (*Figure 5—figure supplement 1*). These results suggest the major functioning of CCKLR-17D3 in Goro-independent nociceptive pathways. We also observed that DSK receptor mutants exhibited more severe thermal phenotypes than *dsk* mutants (*Figure 1*), which may indicate the presence of unidentified DSK- or CCKLR-dependent signaling pathways in nociception-related cells. Further research is apparently necessary to reveal the whole picture of nociceptive regulations mediated by DSK and its receptors in larval *Drosophila*.

## Potential conservation of CCK-mediated descending nociceptive controls

The CCK system is thought to be one of the most ancient neuropeptide systems, suggested to have multiple common physiological functions across taxa (*Elphick et al., 2018*; *Mirabeau and Joly, 2013*; *Jékely, 2013*; *Nässel and Williams, 2014*; *Tinoco et al., 2021*). In this study, we demonstrate that CCKergic signaling in *Drosophila* participates in nociceptive modulation through a descending inhibitory pathway similarly to the mammalian CCK system, adding new evidence about the conserved physiological roles of CCK.

Unlike the peripheral nociceptive systems, it is still challenging to align the *Drosophila* and mammalian CCKergic descending pathways due to low homologies of the CNS structures between *Drosophila* and mammals, wide-spread CCK expression in the mammalian CNS, and the multiple roles of the CCKergic systems in mammalian nociceptive controls (*Heinricher and Neubert, 2004*; *Heinricher et al., 2001*; *Marshall et al., 2012*; *Xie et al., 2005*; *Roca-Lapirot et al., 2019*; *Pommier et al., 2002*; *Noble and Roques, 2002*; *Baber et al., 1989*; *Suh and Tseng, 1990*; *Liu et al., 2018*). However, the common usage of an orthologous molecular pathway in descending controls of nociception between the two evolutionarily distant clades raises a fascinating new hypothesis that the descending control from the brain may also be an ancient, conserved mechanism of nociception, which has emerged in the common ancestor of protostomes and deuterostomes. It will be of interest for future research to investigate whether the role of CCK signaling in descending nociceptive controls is also present in other species.

## Potentials of non-mammalian models to study the mechanisms of pain modulation and pain pathology in the CNS

Non-mammalian model systems have been increasingly recognized as powerful tools to identify novel pain-related molecular pathways (*Burrell, 2017*; *Curtright et al., 2015*; *Milinkeviciute et al., 2012*; *Tobin and Bargmann, 2004*; *Malafoglia et al., 2013*). However, the utilization of these models has so far been mostly limited to the research on peripheral pain pathophysiology, and few studies have used them to investigate central pain pathophysiology (*Burrell, 2017*; *Khuong et al., 2019*). Descending nociceptive control mechanisms are crucial for central pain modulation and have been implicated in the development of chronic pain states in humans (*Chen and Heinricher, 2019*; *Ossipov et al., 2014*). Thus, the current study opens the door to a new approach to using powerful neurogenetic tools and the simpler nervous system of *Drosophila* for elucidating the functional principles of descending nociceptive systems, which may potentially contribute to our understanding of the mechanisms underlying central pain modulation and pain pathology due to dysfunctions of descending modulatory pathways.

## Materials and methods

Fly strains $y^1w^{1118}$ strain was used as the control strain for *dsk*, *CCKLR-17D1*, and *CCKLR-17D3* mutants. *yv; attp2* strain was used for the control of *yv; JF02644* (CCKLR-17D1 RNAi) and *yv; JF02968* (CCKLR-17D3 RNAi). *yw; nos-Cas9/CyO* (NIG-FLY CAS-0011) and *yw; Pr Dr/TM6C Sb Tb* were used for CRISPR/Cas9 mutagenesis. *Dp(3;1)2-2, w^{1118}; Df(3 R)2–2/TM3 Sb* (Bloomington #3688),

*UAS-CCKLR-17D1* (*Chen and Ganetzky, 2012*), and *UAS-CCKLR-17D3* (this study) were used for rescue experiments. *DSK-GAL4* (Bloomington #51981), *DSK-2A-GAL4* (Bloomington #84630), *R69E06-GAL4* (Bloomington #39493), *R69E06-lexA* (Bloomington #54925), *R72F11-lexA* (*Ohyama et al., 2015*), *R70F01-lexA* (Bloomington #53628), *R38A10-lexA* (Bloomington #54106), *R82A10-lexA* (Bloomington #54417), *CCKLR-17D1-T2A-GAL4* (*Kondo et al., 2020*), *CCKLR-17D3-T2A-GAL4* (*Kondo et al., 2020*) and *tsh-GAL80* were used for tissue-specific gene expressions. *40xUAS-IVS-mCD8::GFP* (Bloomington #32195), *10xUAS-IVS-mCD8GFP* (Bloomington #32186), *UAS >CD2 stop >mCD8::GFP hs-flp* (*Wong et al., 2002*), *lexA-rCD2::RFP UAS-mCD8::GFP* (*Ren et al., 2016*), *ppk-CD4::tdGFP* (Bloomington #35842), *UAS-nSyb-GFP$_{1-10}$; lexAop-CD4-GFP$_{11}$* (Bloomington #64314), *UAS-CD4-GFP$_{1-10}$; lexAop-CD4-GFP$_{11}$* (Bloomington #58755), *UAS-Denmark UAS-syt::eGFP* (Bloomington #33065), and *UAS-myrGFP QUAS-mtdTomato::3xHA; trans-Tango* (Bloomington #77124) were used for cellular visualizations. *UAS-GCaMP6m* (Bloomington #42748) was used for GCaMP Ca$^{2+}$ imaging. *UAS-dTRPA1* (Bloomington #26263) (*Hamada et al., 2008*) was used for thermogenetic experiments., *UAS-CaMPARI2* (Bloomington #78316) and *lexAop-CaMPARI2* (Bloomington #78325) were used for CaMPARI2 snapshot imaging of neuronal activity. *dsk$^{sk1}$*, *CCKLR-17D1* deletion mutants, *CCKLR-17D3* deletion mutants, and *UAS-CCKLR-17D3* were generated in this study and described below. *dsk$^{attp}$* strain (*Deng et al., 2019*) was obtained from the Bloomington stock center (#84497). Stocks were kept at 25 °C with 12:12 hr light cycle on a standard food.

## Generating mutants and transgenic lines

Deletion mutants of *dsk*, *CCKLR-17D1*, and *CCKLR-17D3* were generated as described previously (*Kondo and Ueda, 2013*). Briefly, 23 bp-guide RNA (gRNA) sequences specific to the aimed region of the targeted genes were identified using an online tool (http://www.flyrnai.org/crispr/) and 20 bp sequences excluding PAM were cloned to the pBFv-U6.2 vector. Injections of the gRNA-pBFv-U6.2 vectors to yield transgenic fly strains were performed by BestGene Inc. The gRNA-expressing lines were crossed with a nos-Cas9 strain and about 20 independent F1 generations that could potentially possess modifications in the targeted genomic region were established. Lines that had a frameshifting deletion in the targeted region were screened through standard PCR and sequencing. The following gRNA and PCR primer sequences were used to generate *dsk*, *CCKLR-17D1*, and *CCKLR-17D3* mutants:

- *dsk*: GTAGACTAGTCGTCTGCGCT (gRNA), CCTCTAAACACTTGACAGCCGCGGTAACGG (forward primer), and CCGAAACGCATGTGACCGTAGTCATCG (reverse primer).
- *CCKLR-17D1*: GCTTCCGTGATACGCAGACTGGG (gRNA), ATGTGTTTTGTGGATACCCTGT (forward primer), and GGGCTATACCTCCA-TCAGTTTC (reverse primer).
- *CCKLR-17D3*: GCCATATCGGACATGCTGCTGGG (gRNA), GATAGGGA-TGGCTATATGGA CACCGAGC (forward primer), and CTTAGCTGTCCCAATTCCCCCATCTTCT (reverse primer).

The UAS-CCKLR-17D3 line was generated through a ΦC31 integrase–based method. The oligo DNA corresponding to the sequence of CCKLR-17D3 mRNA (+447–2201 of AY231149) was synthesized (Eurofins Genomics) and cloned into the pUASg.attB vector (*Bischof et al., 2013*) using the pENTR/D-TOPO Gateway cloning kit (Thermo Fisher Scientific, MA). The sequence-verified UAS-CCKLR-17D3 construct was integrated into the attP40 site to yield transgenic lines. The injections were performed by BestGene Inc.

## Thermal nociception assay

The experimenters were blinded to genotypes. Larval thermal nociception assays were performed as described previously (*Honjo et al., 2016*), with slight modifications. A custom-made probe with a thermal feedback system was used. Unless otherwise noted, a thermal probe heated to 42 °C was used to detect hypersensitive phenotypes (*Honjo and Tracey, 2018*). The results from all non-responders, defined as individuals that did not exhibit rolling behavior within 10 s, were converted to data points of 11 s for subsequent statistical analyses.

## Thermogenetic activation experiments

The experimenters were blinded to genotypes. A 60 mm dish containing approximately 1 ml distilled water (testing chamber) was placed on a temperature-controlled plate MATS-SPE (TOKAI HIT, Shizuoka, Japan) set at 44.5 °C to equilibrate the water temperature in a testing chamber to 35 ± 1 °C,

which was continuously monitored using a T-type thermocouple wire IT-23 (Physitemp, NJ), USB-TC01 (National Instruments), and the NI Signal Express software (National Instruments, TX). Wandering third instar larvae expressing dTRPA1 by *ppk1.9-GAL4* and/or *DSK-2A-GAL4* were harvested to another 60 mm dish at room temperature (23–25°C), and gently transferred to the 35 °C testing chamber using a paintbrush. All experiments were performed and recorded under a binocular microscope with a camcorder, and the latency from the placement of larvae to rolling was measured offline for each larva.

## Immunohistochemistry

The following antibodies were used in this study: chicken anti-GFP (Abcam, 1:500), mouse anti-GRASP (Sigma #G6539, 1:100) (*Macpherson et al., 2015*), mouse anti-rat CD2 (Bio-Rad, 1:200), rat anti-mCD8 (Caltag, 1:100), rabbit anti-FLRFa (a gift from Dr. Eve Marder, 1:5000) (*Marder et al., 1987*), rabbit anti-CD4 (Novus Biologicals, 1:500), mouse anti-REPO (Developmental Studies Hybridoma Bank 8D12, 1:5), rabbit anti-DsRed (Clontech #632496, 1:200), rat anti-HA (Roche 3F10, 1:100), mouse anti-CaMPARI-red (Absolute antibody 4F6, 1:1000), goat anti-HRP-Cy3 (Jackson ImmunoResearch, 1:100), goat anti-rat Alexa488 (Invitrogen, 1:500), goat anti-chicken Alexa488 (Invitrogen, 1:500), goat anti-mouse Alexa546 (Invitrogen, 1:500), goat anti-rabbit Alexa546 (Invitrogen, 1:500), goat anti-rat Alexa633 (Invitrogen, 1:500), and goat anti-rabbit Alexa633 (Invitrogen, 1:500). Dissected larval tissues were fixed in 4% paraformaldehyde for 30 minutes and then stained as previously described (*Honjo and Tracey, 2018*). The images were acquired by using a Zeiss LSM 510 with a 20 x/0.75 Plan-Apochromat objective or 40 x/1.0 Plan-Apochromat oil immersion objective, Zeiss LSM 700 with a 20 x/0.75 Plan-Apochromat objective or 40 x/1.0 Plan-Apochromat oil immersion objective, or Leica SP5 with a 40 x/0.85 PL APO objective, and digitally processed using Zeiss LSM Image Browser, Leica LAS X Lite, and Adobe Photoshop.

## GRASP and *trans*-Tango experiments

The CD4-GRASP experiments were performed by crossing the *R69E06-lexA DSK-GAL4* strain with the *UAS-CD4-GFP$_{1-10}$; lexAop-CD4-GFP$_{11}$*. The nSyb-GRASP experiments were performed by crossing the *R69E06-lexA DSK-GAL4* with the *UAS-nSyb-GFP$_{1-10}$; lexAop-CD4-GFP$_{11}$*. Wandering 3rd instar larvae were dissected and immunostained as mentioned above. The signals of the reconstituted GFP were detected using a GFP antibody without cross-reaction to split-GFP fragments (Sigma #G6539) (*Macpherson et al., 2015*).

The *trans*-Tango experiments were performed by crossing the *R69E06-lexA, lexAop-rCD2::RFP UAS-mCD8::GFP; DSK-2A-GAL4* with *UAS-myrGFP QUAS-mtdTomato::3xHA; trans-Tango*. Wandering 3rd instar larvae were dissected and immunostained as mentioned above. The following combinations of the primary and secondary antibodies were used to avoid cross-contamination of fluorescent signals in microscopy: chick anti-GFP and anti-chick Alexa488, mouse anti-rCD2 and anti-mouse Alexa546, and rat anti-HA and anti-rat Alexa633.

## FLP-out clone analysis

The *UAS >CD2 stop >mCD8::GFP hs-flp* strain was crossed with *DSK-GAL4* or *DSK-2A-GAL4* to seed vials. Heat-shock induction of FLP-out clones and immunostaining were performed as described previously (*Honjo and Tracey, 2018*). The images of brain samples were acquired and digitally processed as described above.

## GCaMP calcium imaging

Ca$^{2+}$ imaging of Goro neurons using GCaMP6m (*Chen et al., 2013*) was performed as described previously (*Honjo and Tracey, 2018*), with some modifications. Wandering third instar larvae expressing GCaMP6m in Goro neurons by *R69E06-GAL4* were dissected in ice-cold hemolymph-like saline 3.1 (HL3.1) (*Feng et al., 2004*) and imaged in a custom-made imaging chamber containing the HL3.1 equilibrated to the room temperature (23–25°C). A Leica SP8 confocal microscope with resonant scanning system was used to perform three-dimensional time-lapse imaging. Z-stacks consisting of 4–6 optical slices of 512x512 pixel images were acquired at approximately 0.5–1 Hz using a 10 x/0.4 PL APO objective lens with a zoom factor of 8.0. During imaging, a local heat ramp stimulation was applied to the lateral side of the A5 to A7 segment with a custom-made thermal probe. The probe temperature was controlled using a Variac transformer set at 20 V, which generated an approximately

0.6 °C/sec heat ramp stimulus. A T-type thermocouple wire was placed inside of the thermal probe to acquire the probe temperature readings and the data were acquired at 4 Hz through a digitizer USB-TC01 (National Instruments) and the NI Signal Express software (National Instruments). To minimize potential biases caused by day-to-day variations in imaging conditions and achieve fair comparisons, similar numbers of control and experimental genotypes were imaged side-by-side on the same day, using identical microscope settings. To monitor the temperature around the larval CNS during the local heat ramp stimulation, a small thermocouple wire (IT-23, Physitemp) was placed nearby the exposed CNS of semi-dissected larvae.

Maximum intensity projections were generated from the time-series Z-stacks on Leica LAS X Lite and the subsequent analyses of the images and temperature log data were performed using a custom-made code in MATLAB (MathWorks, MA). The region of interest (ROI) was selected as a circular area with a diameter of 15 pixels that covered the neurites of Goro neurons. Cell bodies were not used for the quantification because of small and variable increases in GCaMP6m signals upon heat stimulation. Average fluorescent intensity (F) was calculated for the ROI for each time point. The average of Fs from the first five frames was used as the baseline fluorescent intensity ($F_0$) and percent changes of fluorescent intensity from the baseline [$\Delta F/F_0 = (F - F_0)*100 /F_0$] was calculated for each time point. At least three ROIs were selected from areas that were expected to yield high GCaMP6m signal increases and the ROI that led to the highest peak $\Delta F/F_0$ was chosen for the subsequent statistical analysis. Samples whose highest peak $\Delta F/F_0$ was less than 10% were excluded from the analysis to avoid potential skewness of the data by inclusion of dead/unhealthy samples. Because the images and probe temperatures could not be acquired at the same time, probe temperature for each time point was calculated by linear interpolation from the raw readings. For comparisons among strains, $\Delta F/F_0$ data were binned and averaged in 1 °C intervals.

## CaMPARI2 experiments

Wandering third instar larvae expressing CaMPARI2 with *R69E06-GAL4* or *DSK2A-GAL4* were harvested from vials to a 60 mm petri dish and rinsed with water. Each larva was transferred onto kimwipe to remove moisture and attached with its ventral side up on labeling tape. A custom-made thermal probe (the same probe described for GCaMP experiments above) was placed at the lateral side of the A5 to A7 segment, and a local thermal ramp stimulus from room temperature (23–25°C) to 50 °C was applied using a Variac transformer set at 20 V. Simultaneously with local heat ramp stimulation, 405 nm UV light was applied from 3 cm above the animal using a 405 nm LED stand light (7 mW/cm$^2$, LED405-9VIS, OPTOCODE Corporation, Tokyo, Japan). In 'no heat' control experiments, no voltage was applied to the thermal probe attached to the larva, and only 405 nm light irradiation was applied for 40 seconds (equivalent to the length of time required to apply a temperature ramp stimulus from room temperature to 50 °C with 20 V). In thermogenetic activation experiments, larvae expressing CaMPARI2 under the control of *R69E06-lexA* and dTRPA1 under the control of either only *ppk1.9-GAL4* or both *ppk1.9-GAL4* and *DSK-2A-GAL4* were immobilized on labeling tape with their ventral side up. The larvae were then transferred onto a MATS-SPE temperature-controlled plate set at 35 °C, and simultaneously irradiated with 405 nm light for 10 s from 3 cm above.

Immediately after thermal stimulation and/or photoconversion, the larvae were gently rinsed with water, removed from the labeling tape, and pooled into a new 60 mm dish. Five to fifteen animals were pooled for each experimental group and dissected immediately in ice-cold PBS. The larval CNS samples were immunostained following the protocol described above with mouse anti-CaMPARI red (4F6), rat anti-HA (3F10), anti-mouse Alexa 546, and anti-rat Alexa 633. The images of neurons expressing CaMPARI2 were acquired by using a Zeiss LSM 700 confocal microscope with a 40 x/1.0 Plan-Apochromat oil immersion objective. All parameter settings of the microscope were kept identical across the experimental groups. To enhance green CaMPARI2 fluorescence, a 5% 405 nm laser was mixed with a 20% 639 nm laser. To achieve fair comparisons by minimizing the potential biases from day-to-day variations in stimulation, immunostaining, and imaging, similar numbers of animals from each experimental group were tested, immunostained, and imaged side-by-side on the same day.

Maximum intensity projections of images were generated from the Z-stacks using Zeiss Zen Lite software. Photoconversion rates of CaMPARI2 were quantified using MATLAB. Cell bodies of neurons expressing CaMPARI2 were automatically segmented using Otsu's thresholding algorithm (*Otsu,*

*1979*) based on the anti-HA signal intensity on the blue channel, and each image was cropped around the cell bodies with a margin of 50 pixels to the top, bottom, left, and right. Average intensities of green (green CaMPARI2) and red (anti-CaMPARI red) signals were calculated for cell bodies and the background (cropped image area excluding cell bodies), and the Red/Green ratio was calculated as follows: Red/Green ratio = (Average red signal intensity of cell bodies – Average red signal intensity of the background) / (Average green signal intensity of cell bodies – Average green signal intensity of the background). In the analysis of Goro neurons, images in which both sides of the cell bodies were segmented by Otsu's method were included for the subsequent analyses.

## Data collection and statistical analyses

Mann–Whitney's U-test was used for pair-wise comparisons and Steel's test (the non-parametric equivalent of Dunnet's test) or Steel–Dwass test (the non-parametric equivalent of Tukey-Kramer test) were used for multiple comparisons. Statistical analyses were performed in KyPlot 5.0 and GraphPad Prism 9. The numbers of samples (n) for all experiments indicate the numbers of biological replicates. Each experiment was repeated at least twice on different days to check the reproducibility, and all data were pooled for statistical analyses unless otherwise noted.

## Acknowledgements

We thank the NIG-Fly Stock Center, TRiP at Harvard Medical School (NIH/NIGMS R01-GM084947), the Bloomington Stock Center, and Drs. Masayuki Koganezawa, Takeshi Awasaki, Barry Ganetzky, and Marta Zlatic for fly stocks. We are grateful to Dr. Eve Marder for kindly providing the anti-FLRFa antibody. We also thank Drs. Yoshiki Hayashi, Makoto Hayashi, and Satoru Kobayashi for their support in imaging with Leica SP5 and SP8. This study was supported by grants from JSPS KAKENHI to K Honjo (17K14928, 19K06935, and 22K06328) and HT (17H01378 and 26250001), and grants from the National Center for Geriatrics and Gerontology (21-47), the Uehara Memorial Foundation, and the Mochida Memorial Foundation for Medical and Pharmaceutical Research to K Honjo.

## Additional information

### Funding

| Funder | Grant reference number | Author |
| --- | --- | --- |
| Japan Society for the Promotion of Science | 17K14928 | Ken Honjo |
| Japan Society for the Promotion of Science | 19K06935 | Ken Honjo |
| Japan Society for the Promotion of Science | 22K06328 | Ken Honjo |
| Japan Society for the Promotion of Science | 17H01378 | Hiromu Tanimoto |
| Japan Society for the Promotion of Science | 26250001 | Hiromu Tanimoto |
| National Center for Geriatrics and Gerontology | 21-47 | Ken Honjo |
| Uehara Memorial Foundation | | Ken Honjo |
| Mochida Memorial Foundation for Medical and Pharmaceutical Research | | Ken Honjo |

The funders had no role in study design, data collection and interpretation, or the decision to submit the work for publication.

## Author contributions
Izumi Oikawa, Formal analysis, Validation, Investigation, Visualization, Writing - review and editing; Shu Kondo, Resources, Validation, Writing - review and editing; Kao Hashimoto, Akiho Yoshida, Formal analysis, Investigation; Megumi Hamajima, Investigation; Hiromu Tanimoto, Resources, Funding acquisition, Writing - review and editing; Katsuo Furukubo-Tokunaga, Supervision, Writing - review and editing; Ken Honjo, Conceptualization, Data curation, Formal analysis, Supervision, Funding acquisition, Validation, Investigation, Visualization, Writing - original draft, Writing - review and editing

## Author ORCIDs
Akiho Yoshida http://orcid.org/0000-0001-5733-4894
Megumi Hamajima http://orcid.org/0009-0009-9425-3896
Ken Honjo http://orcid.org/0000-0001-7804-2436

Reviewer #1 (Public Review): https://doi.org/10.7554/eLife.85760.3.sa1
Reviewer #2 (Public Review): https://doi.org/10.7554/eLife.85760.3.sa2
Reviewer #3 (Public Review): https://doi.org/10.7554/eLife.85760.3.sa3
Author Response: https://doi.org/10.7554/eLife.85760.3.sa4

## Additional files

### Supplementary files
• MDAR checklist

### Data availability
All data generated or analyzed during this study are included in the manuscript, supporting file, and source data file.

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
