## [Editor Report · eLife assessment]

This is a very interesting and **important** study that **convincingly** demonstrates a descending pathway for the control of nociception in non-mammalian organisms.

---

## [Referee Report · Reviewer #1 (Public Review)]

The study by Oikawa and colleagues *demonstrates* for the first time that a descending inhibitory pathway for nociception exists in non-mammalian organisms, such as Drosophila. This descending inhibitory pathway is mediated by a *Drosophila* neuropeptide called Drosulfakinin (DSK), which is homologous to mammalian cholecystokinin (CCK). The study creates and uses several *Drosophila* mutants to convincingly show that DSK negatively regulates nociception. They then use several sophisticated transgenic manipulations to demonstrate that a descending inhibitory pathway for nociception exists in *Drosophila*.

Strengths:

This study creates the possibility of using *Drosophila* to study descending nociceptive systems.

CRISPR/Cas9 is used to generate mutants of dsk, CCKLR-17D1, and CCKLR-17D3. The authors then use these mutants to clearly show that DSK negatively regulates nociception.

Several GAL4s are used to clearly show that these effects are likely mediated by two sets of neurons in the brain, MP1 and Sv.

RNAi and rescue experiments further show that CCKLR-17D1, a DSK receptor, functions in Goro neurons to negatively regulate nociception.

Thermogenetic experiments nicely show that activation of DSK neurons attenuates the nociceptive response.

Weaknesses:

Future studies should address how DSK negatively regulates nociception. An earlier study at the *Drosophila* nmj shows that loss of DSK signaling impairs neurotransmission and synaptic growth. In the current study, loss of CCKLR-17D1 in Goro neurons seems to increase intracellular calcium levels in the presence of noxious heat. An interesting future study would be the examination of the underlying mechanisms for this increase in intracellular calcium.

---

## [Referee Report · Reviewer #2 (Public Review)]

This is an exceptional study that provides conclusive evidence for the existence of a descending pathway from the brain that inhibits nociceptive behavioral outputs in larvae of *Drosophila melanogaster*. The authors identify both molecular and neuronal/cellular components of this pathway. Converging lines of evidence and conclusive genetic experiments indicate that the neuropeptide, drosulfakinin (DSK), and its receptors (CCK1 and CCK2) function to inhibit nociception behaviors. Interestingly, the authors show that the relevant DSK neurons have cell bodies that are in the larval brain and that these neurons send projections into the thoracic ganglion and ventral nerve cord. Several lines of evidence support the hypothesis that fourth-order nociceptive neurons called Goro, are one relevant target for these outputs. RNAi knockdown of the CCK1 receptor in these cells sensitizes behavioral and physiological responses to noxious heat. Second, the axons of DSK neurons form physical contact with processes of Goro neurons as revealed by GRASP analysis. However, the authors' careful experiments indicate that the contacts between axons and Goro neurites might not be indicative of direct synapses and instead might operate through the bulk transmission of the peptidergic signals. The study raises many interesting questions for future study such as what behavioral contexts might depend on this pathway. Using the CAMPARI approach, the authors do not find that the DSK neurons are activated in response to nociceptive input but instead suggest that these cells may be tonically active in gating nociception. Future studies may find contexts in which the output of the DSK neurons is inhibited to facilitate nociception or contexts in which the cells are more active to inhibit nociception.

---

## [Referee Report · Reviewer #3 (Public Review)]

This study describes a descending circuit that can modulate pain perception in the *Drosophila larvae*. While descending inhibition is a major component of mammalian pain perception, it is not known if a similar circuit design exists in fruit flies. Overall the authors use clean logic to establish a role for DSK and its receptor in regulating nociception. The following concerns still stand:

It's not completely clear why the authors are staining animals with an FLRFa antibody. Can the authors stain WT and DSK KO animals with a DSK antibody? Also, can the authors show in supplemental what antigen the FLRFa antibody was raised against, and what part of that peptide sequence is retained in the DSK sequence? This overall seems like a weakness in the study that could be improved on in some way by using DSK-specific tools.What is the phenotype of DSK-Gal4 x UAS-TET animals? They should be hyper-reactive. If it's lethal maybe try an inducible approach.Figure 9. This was not totally clear, but I think the authors were evaluating spontaneous (i.e. TRPA1-driven) rolling at 35C. The critical question is "Does activating DSK-expressing neurons suppress acute heat nociception?" and this hasn't really been addressed. The inclusion of PPK Gal4 + DSK Gal4 in the same animal clouds the overall conclusions the reader can draw. The essential experiment is to express UAS-dTRPA1 in DSK-Gal4 or GORO-Gal4 cells, heat the animals to ~29C, and then test latency to a thermal heat probe (over a range of sub and noxious temperatures). Basically, prove the model in Figure 10 showing ectopic activation or inhibition for each major step, then test heat probe responses.It would also then be interesting to see how strong the descending inhibition circuit is in the context of UV burn. If this is a real descending circuit, it should presumably be able to override sensitization after injury.

---

## [Author Response]

The following is the authors' response to the original reviews.

We sincerely thank all the editors and reviewers for taking the time to evaluate this study. Here is our point-by-point response to the reviewers’ comments and concerns.

**Reviewer #1 (Public Review):**The study by Oikawa and colleagues demonstrates for the first time that a descending inhibitory pathway for nociception exists in non-mammalian organisms, such as *Drosophila*. This descending inhibitory pathway is mediated by a *Drosophila* neuropeptide called Drosulfakinin (DSK), which is homologous to mammalian cholecystokinin (CCK). The study creates and uses several *Drosophila* mutants to convincingly show that DSK negatively regulates nociception. They then use several sophisticated transgenic manipulations to demonstrate that a descending inhibitory pathway for nociception exists in *Drosophila*.[…]Weaknesses:A minor weakness in the study is that it is unclear how DSK negatively regulates nociception. An earlier study at the *Drosophila* nmj shows that loss of DSK signaling impairs neurotransmission and synaptic growth. In the current study, loss of CCKLR-17D1 in Goro neurons seems to increase intracellular calcium levels in the presence of noxious heat. An interesting future study would be the examination of the underlying mechanisms for this increase in intracellular calcium.

We thank the reviewer for the kind and very positive evaluation of our manuscript. We agree that this study has not elucidated the intracellular molecular pathway(s) downstream of CCKLR-17D1 that are involved in the regulation of the activity of Goro neurons, and we think that it would definitely be an interesting topic for future research.

**Reviewer #1 (Recommendations For The Authors):**
The response latencies for the control yw larvae seem large, with many larvae appearing to be insensitive to the thermal stimulus. Is this just an effect of the yw genetic background? A brief discussion of this might be helpful.

We thank the reviewer for pointing this out. We have also noticed that the *yw* control larvae tend to show longer response latencies than the other control strains, and in the revised manuscript, we have added the following sentence in the Result section (Lines 91–94):

“We have noticed that the *yw* control strain, which was used by us to generate the *dsk* and receptor deletion mutants, showed relatively longer response latencies to the 42 °C probe compared to the other control strains used in this study. This may be attributed to the effect of the genetic background, although, presently, the cause for this difference is unknown.”

**Reviewer #2 (Public Review):**_This is an exceptional study that provides conclusive evidence for the existence of a descending pathway from the brain that inhibits nociceptive behavioral outputs in larvae of *Drosophila melanogaster*. […] The study raises many interesting questions for future study such as what behavioral contexts might depend on this pathway. Using the CAMPARI approach, the authors do not find that the DSK neurons are activated in response to nociceptive input but instead suggest that these cells may be tonically active in gating nociception. Future studies may find contexts in which the output of the DSK neurons is inhibited to facilitate nociception, or contexts in which the cells are more active to inhibit nociception._**Reviewer #2 (Recommendations For The Authors):**I have no recommendations for the authors as this is a very complete and thoroughly executed study. The writing is crystal clear.

We thank the reviewer for the kind and very positive evaluation of our manuscript. We are happy to know that our current manuscript was deemed to be clear and convincing by the reviewer.

**Reviewer #3 (Public Review):**[…] Overall the authors use clean logic to establish a role for DSK and its receptor in regulating nociception. I have made a few suggestions that I believe would strengthen the manuscript as this is an important discovery.Major comments:1. It's not completely clear why the authors are staining animals with an FLRFa antibody. Can the authors stain WT and DSK KO animals with a DSK antibody? Also, can the authors show in supplemental what antigen the FLRFa antibody was raised against, and what part of that peptide sequence is retained in the DSK sequence? This overall seems like a weakness in the study that could be improved on in some way by using DSK-specific tools.

We thank the reviewer for this query. We would like to clarify that we first tried the FLRFa antibody to visualize an RFamide-type neuropeptide other than DSK in *Drosophila* and found that the staining pattern is quite similar to that of anti-DSK, as shown by Nichols et al. [1]. According to the original paper describing the anti-FLRFa antisera [2] (already cited in the reviewed manuscript), the antigen used to raise it was the Phe-Met-Arg-Phe-NH2 peptide conjugated with succinylated thyroglobulin, and the study experimentally shows that the antibody well binds to peptides containing Met-Arg-Phe-NH2 or Leu-Arg-Phe-NH2 sequence and has 100% cross-reactivity to FLRFa. As DSK contains Met-Arg-Phe-NH2 sequence [3], the cross-reaction of this antibody to DSK is consistent with the description of the original study.

Although we were unable to use an antibody specific to DSK, our staining data with *dsk* deletion mutants and the expression pattern of DSK-2A-GAL4 corroborate each other (Figure 2 and Figure 2-figure supplement 1), which we believe provides compelling evidence for the specific expression of DSK in MP1 and Sv neurons, and for that DSK-2A-GAL4 is a reasonably effective tool to specifically manipulate DSK-expressing neurons.

2. What is the phenotype of DSK-Gal4 x UAS-TET animals? They should be hyper-reactive. If it's lethal maybe try an inducible approach.

We thank the reviewer for this question. Unfortunately, we have not attempted this experiment, although we agree that this would be a nice addition to further strengthen the study if TET worked well in the DSKergic neurons.

3. Figure 9. This was not totally clear, but I think the authors were evaluating spontaneous (i.e. TRPA1-driven) rolling at 35C. The critical question is "does activating DSK-expressing neurons suppress acute heat nociception" and this hasn't really been addressed. The inclusion of PPK Gal4 + DSK Gal4 in the same animal kind of clouds the overall conclusions the reader can draw. The essential experiment is to express UAS-dTRPA1 in DSK-Gal4 or GORO-Gal4 cells, heat the animals to ~29C, and then test latency to a thermal heat probe (over a range of sub and noxious temperatures). Basically prove the model in Figure 10 showing ectopic activation or inhibition for each major step, then test heat probe responses.

We thank the reviewer for suggesting ideas for alternative experiments to potentially strengthen our conclusion. Regarding experiments using heat probes, previous studies have demonstrated that (i) Blocking ppk1.9-GAL4-positive C4da neurons almost completely abolishes the larval nociceptive response to local heat stimulations [4]; (ii) Local heat stimuli above 39 °C readily activate C4da neurons and larval nociceptive rolling [5-9]; and (iii) Thermogenetically or optogenetically activating these neurons is sufficient to trigger Goro neurons and larval rolling [4, 10-12]. Thus, it has now been made clear that heat probes induce larval nociceptive rolling via excitation of the C4da pathway, and we believe that our experiments using thermogenetic activation of C4da neurons can be safely interpreted as an alternative to experiments using heat probes. Using heat probes demands a more complicated experimental set-up to be combined with CaMPARI imaging experiments, and this is another reason why we preferred to take the thermogenetic approach.

We have also considered the experiment using Goro-GAL4 instead of ppk-GAL4. However, if dTRPA1 artificially activates Goro neurons far downstream of the neuronal mechanism by which MP1 activation suppresses Goro neuron activity, the effect of MP1 activation may be bypassed and masked. As we currently do not know the epistasis between dTRPA function and the effect of MP1 activation in modulating the activity of Goro neurons, we rather chose to activate C4da neurons by using ppk-GAL4, which likely resulted in more natural activation of Goro neurons than dTRPA1-triggered direct activations.

4. It would also then be interesting to see how strong the descending inhibition circuit is in the context of UV burn. If this is a real descending circuit, it should presumably be able to override sensitization after injury.

Indeed, this is an interesting avenue to explore in future studies to understand the type of situation in which the DSKergic descending system functions to control nociception.

**Reviewer #3 (Recommendations For The Authors):**Overall this is a good story and the claims are generally supported with experimental evidence. The way to really improve this study would be to use more precise and definitive tools, like specific antibodies, specifically targeted genes, and better temporal control of the descending circuit to prove this is inducible sufficient to suppress acute thermal nociception and this occurs only via a descending pathway, etc. However this would be exponentially more work, and so the authors I guess need to weigh the cost-benefit of definitive proof vs. strong evidence for their claims. Overall I think this study will be the beginning of a new line of inquiry in the field that has the potential to guide our understanding also of mammalian descending pathways, and as such, this study is of value to the community.

We appreciate the reviewer’s multiple interesting ideas for experiments that could have been performed to further reinforce our findings. We agree that some experiments that the reviewer suggested would potentially strengthen this work if supplemented. However, as aforementioned, in our humble opinion, we think that the experiments that the reviewer suggested are either outside the scope of this paper or have no significant benefits over the experiments that were already conducted, and hence are not essential to the present study.

**References**

Nichols, R. and I.A. Lim, *Spatial and temporal immunocytochemical analysis of drosulfakinin (Dsk) gene products in the Drosophila melanogaster central nervous system.* Cell Tissue Res, 1996. **283**(1): p. 107-16.Marder, E., et al., *Distribution and partial characterization of FMRFamide-like peptides in the stomatogastric nervous systems of the rock crab, Cancer borealis, and the spiny lobster, Panulirus interruptus.* J Comp Neurol, 1987. **259**(1): p. 150-63.Nassel, D.R. and M.J. Williams, *Cholecystokinin-like peptide (DSK) in Drosophila, not only for satiety signaling.* Front Endocrinol, 2014. **5**.Hwang, R.Y., et al., *Nociceptive neurons protect Drosophila larvae from parasitoid wasps.* Curr Biol, 2007. **17**(24): p. 2105-2116.Tracey, W.D., Jr., et al., *painless, a Drosophila gene essential for nociception.* Cell, 2003. **113**(2): p. 261-73.Xiang, Y., et al., *Light-avoidance-mediating photoreceptors tile the Drosophila larval body wall.* Nature, 2010. **468**(7326): p. 921-6.Burgos, A., et al., *Nociceptive interneurons control modular motor pathways to promote escape behavior in Drosophila.* eLife, 2018. **7**.Honjo, K. and W.D. Tracey, Jr., *BMP signaling downstream of the Highwire E3 ligase sensitizes nociceptors.* PLoS Genet, 2018. **14**(7): p. e1007464.Im, S.H., et al., *Tachykinin acts upstream of autocrine Hedgehog signaling during nociceptive sensitization in Drosophila.* eLife, 2015. **4**: p. e10735.Ohyama, T., et al., *A multilevel multimodal circuit enhances action selection in Drosophila.* Nature, 2015. **520**(7549): p. 633-9.Honjo, K., R.Y. Hwang, and W.D. Tracey, Jr., *Optogenetic manipulation of neural circuits and behavior in Drosophila larvae.* Nat Protoc, 2012. **7**(8): p. 1470-8.Zhong, L., et al., *Thermosensory and non-thermosensory isoforms of Drosophila melanogaster TRPA1 reveal heat sensor domains of a thermoTRP channel.* Cell Rep, 2012. **1**(1): p. 43-55.